



# A regional modelling study of halogen chemistry within a volcanic plume of Mt Etna's Christmas 2018 eruption

Herizo Narivelo [1], Paul David Hamer [2], Virginie Marécal [1], Luke Surl [3,4,7], Tjarda Roberts [3,5], Sophie Pelletier [1], Béatrice Josse [1], Jonathan Guth [1], Mickaël Bacles [1], Simon Warnach [6], Thomas Wagner [6], Stefano Corradini [8], Giuseppe Salerno [9], and Lorenzo Guerrieri [8]

[1]Centre National de Recherches Météorologiques, Université de Toulouse, Météo-France, CNRS, Toulouse, 31000, France
[2]NILU – Norwegian Institute for Air Research, P.O. Box 100, Kjeller, 2027, Norway
[3]Laboratoire de Physique et Chimie de l'Environnement et de l'Espace, UMR7328, CNRS-Université d'Orléans, 45000, France
[4]Laboratoire Atmosphères, Milieux, Observations Spatiales/Institut Pierre Simon Laplace (LATMOS/IPSL), Sorbonne Université, UVSQ, CNRS, Paris, France
[5]LMD/IPSL, ENS, Université PSL, École Polytechnique, Institut Polytechnique de Paris, Sorbonne Université, CNRS, Paris France
[6]Max-Planck-Institut für Chemie (MPI-C), Mainz, Germany
[7]Department of Biosciences, University of Exeter, Exeter, United Kingdom
[8]Istituto Nazionale di Geofisica e Vulcanologia, INGV, ONT, 00143 Roma, Italy
[9]Istituto Nazionale di Geofisica e Vulcanologia, INGV, Osservatorio Etneo, Italy

**Correspondence:** Paul Hamer (paul.hamer@nilu.no)

**Abstract.** Volcanoes are known to be important emitters of atmospheric gases and aerosols, which for certain volcanoes can include halogen gases and in particular HBr. HBr emitted in this way can undergo rapid atmospheric oxidation chemistry (known as the bromine-explosion) within the volcanic emission plume leading to the production of bromine oxide (BrO) and ozone depletion. In this work, we present the results of a modelling study of a volcanic eruption from Mt Etna that occurred

around Christmas 2018 that lasted 6 days. The aims of this study are to demonstrate and evaluate the ability of the regional 3D Chemistry Transport Model MOCAGE to simulate the volcanic halogen chemistry in this case study, to analyse the variability of the chemical processes during the plume transport, and to quantify its impact on the composition of the troposphere at a regional scale over the Mediterranean basin.

The comparison of the tropospheric $SO_2$ and BrO columns from 25 to 30 December 2018 from the MOCAGE simulation with

the columns derived from the TROPOMI satellite measurements shows a very good agreement for the transport of the plume and a good consistency for the concentrations if considering the uncertainties in the flux estimates and the TROPOMI columns. The analysis of the bromine species' partitioning and of the associated chemical reaction rates provides a detailed picture of the simulated bromine chemistry throughout the diurnal cycle and at different stages of the volcanic plume's evolution. The partitioning of the bromine species is modulated by the time evolution of the emissions during the 6 days of the eruption, by

the meteorological conditions, and by the distance of the plume from the vent/the time since the emission. As the plume travels further from the vent, the halogen source gas HBr becomes depleted, BrO production in the plume becomes less efficient, and ozone depletion (proceeding via the Br + $O_3$ reaction followed by the BrO self-reaction) decreases. The depletion of HBr





relative to the other prevalent hydracid HCl leads to a shift in the relative concentrations of the Br$^-$ and Cl$^-$ ions, which in turn leads to reduced production of Br$_2$ relative to BrCl.

The MOCAGE simulations show a regional impact of the volcanic eruption on the oxidants OH and O$_3$ with a reduced burden of both gases that is caused by the chemistry in the volcanic plume. This reduction in atmospheric oxidation capacity results in a reduced CH$_4$ burden. Finally, sensitivity tests on the composition of the emissions carried out in this work show that the production of BrO is higher when the volcanic emissions of sulfate aerosols are increased but occurs very slowly when no sulfate and Br radicals are assumed to be in the emissions. Both sensitivity tests highlight a significant impact on the oxidants in the troposphere at the regional scale of these assumptions.

All the results of this modelling study are consistent with the previous studies carried out on the volcanic halogens modelling.

## 1 Introduction

Volcanoes are known to be significant emitters of atmospheric gases and aerosols, both through explosive eruption and persistent quiescent degassing (von Glasow et al., 2009). The main species emitted by volcanoes are H$_2$O (50%-90%) and CO$_2$ (1%-40%), species that are already very abundant in the atmosphere. Sulphur compounds are also observed in volcanic plumes mainly in the form of SO$_2$ species. SO$_2$ is an important species in the atmosphere because it becomes oxidised to sulfate particles and has an impact on air quality, the environment, climate and human health (Bluth et al., 1993; Kelly, 1997; Grainger and Highwood, 2003; Robock, 2000). Volcanoes also emit halogen compounds typically in smaller quantities than SO$_2$ and are in the reduced hydracid form HX where X is either Br, Cl, F or I (Gerlach, 2004; Textor et al., 2004; Gutmann et al., 2018).

During a volcanic emission, the magmatic air that comes out of the crater is at a very high temperature (> 500°C). A mixing of magmatic air and atmospheric air occurs at very high temperatures at the vent that is believed to lead to the formation of Br, Cl, H, OH radicals, possibly NO (Roberts et al. (2019) and references therein). In this effective source region a significant amount of sulfate aerosols are formed, called "primary" sulfate as opposed to the secondary sulfate aerosols formed from SO$_2$ later in the plume. This mixture changes the chemical composition of the original magmatic gas emissions. After, the plume cools down rapidly to a temperature close to that of the atmosphere, and chemical processes occur that lead to the production of other halogen species within the plume in the following minutes to hours. Chemically formed bromine oxide (BrO) was detected in volcanic plumes firstly in the La Soufrière volcanic plume at Montserrat (Bobrowski et al., 2003) by Differential Optical Absorption Spectroscopy (DOAS). Since then, many other observations of BrO have been made in other volcanic plumes (Hörmann et al., 2013), such as at Mt Etna in Sicily (Oppenheimer et al., 2006), Villarica in Chile (Bobrowski et al., 2007), Erebus in Antarctica (Boichu et al., 2011), Kasatochi in Alaska (Theys et al., 2009) and Eyjafjallajökull (Heue et al., 2011).

In volcanic plumes, the process producing BrO from HBr in large quantities and leading to a loss of ozone is called the bromine-explosion cycle (e.g. Oppenheimer et al., 2006; Bobrowski et al., 2007; Roberts et al., 2009; Gutmann et al., 2018). This is an auto-catalytic cycle by which one molecule of BrO forms two BrO from the loss of one molecule HBr and of two molecules of ozone. The cycle also requires the presence of HO$_x$ and acidic aerosols to support its continuation. Sulfate aerosols play





a major role since they provide the acidic particles for the heterogeneous reactions that are crucial in the bromine-explosion cycle.

Previous numerical modelling studies based on the PlumeChem and MISTRA 0D/1D models have mainly focused on the chemistry in the volcanic plume at local and short time scales (a few hours maximum) close to the vent (e.g., Bobrowski et al., 2007; Roberts et al., 2009; Kelly et al., 2013; Roberts et al., 2014, 2018; Surl et al., 2015). Results of these modelling studies conclude that the main cause of the depletion of ozone within the plume is from the reactive bromine generated through the auto-catalytic reactions. The presence of the $HO_x$, $NO_x$, and the primary sulfate aerosols accelerates the processes of BrO production and thus of ozone depletion. Most of the previous modelling studies were initialised using the outputs of the thermodynamic equilibrium software (HSC) that was used to represent the high temperature processes at vent. Nevertheless, recent studies (e.g., Martin et al., 2012; Roberts et al., 2019; Kuhn et al., 2022) have shown that the chemistry at very high temperature at the vent cannot be captured by the thermodynamic equilibrium model assumptions used in HSC, especially for NOx. Even if the processes that take place in these conditions are not yet fully understood, it is clear that some $HO_x$ oxidants and Br radicals are necessary at this early stage to "kick-start" the onset of bromine-explosion.

To understand the effects of the halogen plume chemistry on the air composition further from the vent, a 3D modelling approach is required to represent the 3D transport and mixing together with the chemical evolution of the plume. So far only two 3D studies modelling halogen chemistry in volcanic plumes have been carried out (Jourdain et al., 2016; Surl et al., 2021). Surl et al. (2021) used the 3D WRF-Chem Volcano (WCV) model with a resolution of 1 km and focused on the passive degassing plume of Mt Etna volcano (Sicily, Italy) during the summer of 2012 and examined its fate up to several tens of kilometres downwind. Their reference simulation showed that the increase in BrO and the associated ozone depletion in the plume are consistent with the observations. In addition, their sensitivity simulations validated the relevance of assuming that near-vent radicals are produced from high temperature chemistry to rapidly trigger the halogen cycle in the plume. To date, Jourdain et al. (2016) is the only 3D modelling study to analyse a the impact of halogens in a volcanic eruption on a regional scale, regional meaning here with a domain greater than 500 x 500 $km^2$. They simulated the reactive halogen plume for the case study of the extreme passive degassing event of the Ambrym volcano, Vanuatu that occurred in early 2005 for which tropospheric columns of $SO_2$ and BrO from DOAS measurements were available in the plume at a distance between 15 and 40 km from the vent. The Coupled Chemistry Aerosol-Tracer Transport model to the Brazilian developments on the Regional Atmospheric Modelling System (CCATT-BRAMS, Longo et al. (2013)) was used with nested grids from 50 km down to 0.5 km horizontal resolutions to capture the plume processes at the small scale but also to simulate the tropospheric impact of the dispersed plume at the regional scale. The results showed the influence of the volcanic emissions at the local and regional scales on the main oxidants in the atmosphere (depletion of $HO_x$, $NO_x$ and as well as of ozone) and the increase of the lifetime of $CH_4$.

A step further is to analyse and to quantify the effect of volcanic halogens on the air composition at the regional scale for other case studies of halogen-rich emitting volcanoes and even to go further up to the global scale. For this purpose, it is possible to use a regional or global 3D atmospheric chemistry model. Here, we propose to run 3D simulations with the MOCAGE model, which is a chemistry-transport model (CTM) (Guth et al., 2016; Cussac et al., 2020; Lamotte et al., 2021) that can be used at regional and global scales, to study an eruption of the Mt Etna volcano that lasted several days around Christmas 2018. In



this framework, the main objectives of the paper are to test the capability of the 3D MOCAGE regional CTM with a 0.2° x 0.2° horizontal resolution to simulate the bromine-explosion cycle on this case study, to analyse the variability of the chemical processes in the volcanic plume at different distances from the vent, and to quantify the regional impact of this eruption on tropospheric composition over the whole Mediterranean basin. The present paper is different from the regional study carried out by Jourdain et al. (2016). Jourdain et al. (2016) focused on the passive degassing of the Ambrym volcano in the tropical Pacific region while our paper studies an eruption of the Mt Etna volcano located in the mid-latitudes. Moreover, Jourdain et al. (2016) used several nested grids with its finest horizontal resolution down to 0.5 km in contrast to the 3D MOCAGE CTM simulation whose domain has a coarser horizontal resolution ($\sim$22 km x $\sim$18 km around Mt Etna's latitude).

The present study builds on Marécal et al. (2022) which used a 1D vertical profile version of MOCAGE to prepare the 3D-MOCAGE simulations of halogens in volcanic plumes. For this, they completed MOCAGE chemistry scheme with the reactions needed for the bromine-explosion cycle. Their results show that the 1D MOCAGE model is able to produce the bromine-explosion cycle with values consistent with observations and with previous modelling studies.

This paper is organised as follows. First, we present in section 2 the description of the case study of the Mt Etna eruption event around Christmas 2018. We also present in this section the TROPOspheric Atmosphere Monitoring Instrument (TROPOMI) satellite measurements which captured the $SO_2$ and BrO signature of the volcanic plume during the eruption time period. In section 3, we present the general description of the 3D MOCAGE model, the halogen plume chemistry as well as the setup of the simulations. In section 4, we present and discuss the simulation results. The conclusions are presented in section 5.

## 2 Description of the case study: Mt Etna eruption event around Christmas 2018

### 2.1 General description

Mt Etna is a stratovolcano with several craters located on the island of Sicily, Italy with an altitude of about 3330 m above sea level. It is one of the most active volcanoes in the world and the most active in Europe (e.g. Bonaccorso et al., 2004; Calvari and Nunnari, 2022), and one of the most important sources of $SO_2$ during and between eruptions (e.g. Allard et al., 1991; Burton et al., 2003; Aiuppa et al., 2008; Carn et al., 2017). Its emission type is mainly passive degassing, but since 2011 there have been more eruptive phases (from strombolian eruptions to lava fountains) (e.g. Calvari and Nunnari, 2022). It is also known as an important source of halogen trace gases (e.g Aiuppa et al., 2005, 2008; Oppenheimer et al., 2006).

The case study of the eruptive period around Christmas 2018 (e.g. Calvari et al., 2020; Paonita et al., 2021) was chosen for several reasons. Firstly, this eruption injected significant amounts of $SO_2$ into the troposphere on 24 December and for the 6 days after. Secondly, the $SO_2$ plume and associated BrO were captured by the TROPOMI satellite measurements everyday over the whole eruption period (see detail in sub-section 2.3). Thirdly, the $SO_2$ emissions of the eruption are well documented (Corradini et al., 2020, 2021). In these studies, various ground-based and space-borne instruments were used to estimate the different parameters of the eruption (plume height, $SO_2$ concentration, $SO_2$ emission flux). And finally, we benefit from Lamotte et al. (2023) whose study was focused on the analysis of MOCAGE simulations of $SO_2$ for the Christmas 2018





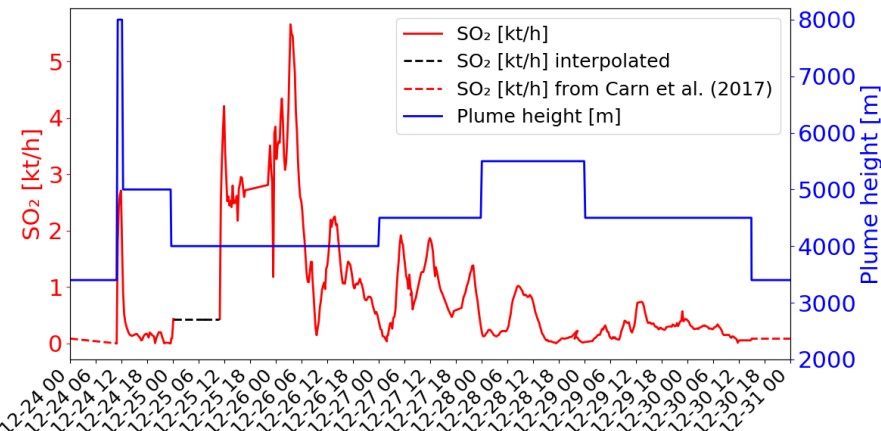

**Figure 1.** SO$_2$ fluxes in kt/h (red colour) and top altitude in m (blue colour) of the emissions as a function of time from the data of Corradini et al. (2020, 2021). The dashed lines correspond to the time periods when data is missing and give the values chosen for the simulations (see explanations in section 3.3.2).

eruption. They used two estimates of SO$_2$ emissions from different instruments in MOCAGE and evaluated the simulated plume against SO$_2$ column retrieved from several space-borne instruments.

### 2.2 SO$_2$ volcanic emissions

Before the Christmas eruption, there was a period of moderate explosive activity and small lava flows (e.g., Laiolo et al., 2019; Calvari et al., 2020). Eruptive activity escalated on 24 December 2018, involving the opening of an eruptive fissure in the summit south-east flank of the volcano, and accompanied by a seismic swarm and shallow earthquakes and by large and widespread ground deformation (e.g Bonforte et al., 2019; Calvari et al., 2020). This led to an eruptive plume that was injected into the upper troposphere at 11:15 UTC. The eruptive period lasted until 30 December. Corradini et al. (2020, 2021) used several instruments to estimate the SO$_2$ flux for the period of the eruption. In our study, we select the fluxes of SO$_2$ estimated from the SEVIRI (Spinning Enhanced Visible and Infrared Imager) satellite instrument. Compared to the other types of measurements, SEVIRI offers a larger temporal coverage and a higher frequency of SO$_2$ emission flux estimations (15 minutes from 24 December to 30 December) (Corradini et al., 2021). In addition, Lamotte et al. (2023) showed that the SEVIRI estimates give better results compared to the SO$_2$ emission flux estimated from the ground-based instrument FLAME (FLux Automatic MEasurement) (Salerno et al., 2009, 2018).

The temporal variation of the emission flux from SEVIRI (Corradini et al., 2021) is shown in Fig. 1 (red line). Corradini et al. (2021) estimated that the total uncertainty of the SO$_2$ flux is approximately 45%. The top height of the volcanic emissions are also presented in Fig. 1 (blue line). The plume height is estimated from SEVIRI measurements on 24 of December at the maximum of the eruption (8000 m at 11:15 UTC). For the other days (25 to 30 of December), the top height was obtained from the ground-based calibrated VIS cameras network in the Mt Etna area (more detail in Corradini et al. (2020)). Thought



the eruption was not marked by an exceptional plume height (maximum of ~8 km altitude) (Corradini et al., 2018), active $SO_2$-degassing was important. Remarkable changes at both temporal and magnitude scales, with values exceeding the typical degassing of Mt Etna during quiescent conditions (Salerno et al., 2018) were measured by the ground FLAME DOAS network

since November2018. $SO_2$ flux steadily increased to then climax in late December 2018 (Paonita et al., 2021).

## 2.3    Transport of the volcanic plume

The plume from this eruption has been well sampled by the TROPOMI space-borne sensor. TROPOMI is a nadir-viewing passive imaging spectrometer onboard the Copernicus Sentinel-5 Precursor satellite. The resolution of the instrument is 3.5 km x 7.5 km in exact nadir direction for measurements taken before 6 August 2019. The temporal resolution over Mt Etna is 1

overpass per day (around 11:00 – 12:00 UTC) except on 27 December when there were 2 overpasses.

The tropospheric columns of $SO_2$ and BrO retrieved in the volcanic plume from the TROPOMI satellite observations around Christmas 2018 (from 25 to 30 December) obtained using a retrieval algorithm based on the DOAS method (Differential Optical Absorption Spectroscopy) (Hörmann et al., 2013; Warnach, 2022) are presented in Fig. 2. The $SO_2$ and BrO uncertainties are estimated at 35%. The $SO_2$ column for 24 December are not shown because the TROPOMI overfly was very close to the

beginning of the eruption and thus only captured the plume on a few pixels.

On 25 December, TROPOMI observations show that the $SO_2$ volcanic plume is transported eastwards and crossed the Mediterranean basin, passing over Greece and Cyprus. From 26 December, the $SO_2$ volcanic plume changed to a more southward directions towards north Africa due to the evolution of the meteorological conditions. Figure 2 also shows the BrO columns in the volcanic plume from 25 to 30 December estimated from TROPOMI satellite measurements. The tropospheric column

value of $SO_2$ is much higher than that of BrO since the emission of bromine compounds is much lower than those of $SO_2$ emissions. We note that there is a good agreement in the location of the plume and the relative intensity between BrO and $SO_2$.

## 3    MOCAGE CTM general description and setup of the simulations

### 3.1    MOCAGE model overview

MOCAGE is a three-dimensional chemistry transport model developed at the Centre National de Recherches Météorologiques

of Météo-France. The model aim is to describe the chemical state of the atmosphere from the surface to the mid-stratosphere. The MOCAGE model is used in different fields of research such as the study of the climatic impact on atmospheric composition (e.g., Lamarque et al., 2013; Lacressonnière et al., 2014) or of the upper troposphere and lower stratosphere gaseous composition (e.g., Barré et al., 2014; Cussac et al., 2020). The model is also used for operational purposes for air quality forecasts for the PREV'AIR program (Rouil et al., 2009) for France and in the framework of the Copernicus project for Europe

(Marécal et al., 2015), and for monitoring volcanic eruptions as part of the Toulouse VAAC (Volcanic Ash Advisory Center) of Météo-France which is responsible for an area including part of Europe and Africa.





**Figure 2.** Tropospheric column of $SO_2$ (left column) and BrO (right column) in molec.cm$^{-2}$ retrieved from TROPOMI satellite measurements from 25 to 30 December 2018.



### 3.1.1 Model geometry and meteorological parameters

MOCAGE CTM can be used with global and/or regional domains thanks to its grid-nesting capability. Each external domain is used as boundary conditions for the internal domains at their edges. The vertical grid has 47 levels from the surface to 5hPa (about 35km above sea level), with 7 levels in the planetary boundary layer, 20 in the free troposphere and 20 in the stratosphere. The vertical levels are expressed in sigma hybrid coordinates, meaning that the model levels closely follow the topography near the surface and follow the pressure levels in the upper atmosphere.

As MOCAGE is an off-line model, it does not resolve the meteorological parameters. The meteorological variables used as input (pressure, temperature, wind, moisture, cloud and precipitation) are obtained from a meteorological model (e.g. the operational numerical weather prediction model Météo-France-ARPEGE or ECMWF-IFS) or a climate model (e.g. CNRM-CM climate model).

### 3.1.2 Gaseous chemistry

The MOCAGE chemical scheme is called RACMOBUS. It merges two chemical schemes: RACM (Stockwell et al., 1997) in the troposphere and REPROBUS (Lefèvre et al., 1994) in the stratosphere. Additionally, several reactions have been included to complete the sulphur cycle in the troposphere (Guth et al., 2016). The RACM scheme represents the tropospheric chemistry of organic compounds. Halogen chemistry is not present in the RACM tropospheric scheme: halogen reactions are added to the model in this study (Section 3.2). The REPROBUS scheme is intended to describe the evolution of stratospheric ozone and therefore includes the reactions that take place in the stratosphere, in particular with halogen species. In the original version of the MOCAGE chemistry scheme, 112 species and 379 chemical reactions (photolysis, gaseous and heterogeneous reactions) are taken into account.

### 3.1.3 Aerosols

The primary and secondary aerosols are represented in MOCAGE model (Martet et al., 2009; Sič et al., 2015; Guth et al., 2016; Descheemaecker et al., 2019). The primary aerosols are composed of 4 species: black carbon, primary organic carbon, sea salt and desert dust. The secondary inorganic aerosols (SIAs) are composed of 3 species, sulfate, nitrate and ammonium, and are implemented in MOCAGE (Guth et al., 2016). The latest version of the thermodynamic equilibrium model ISORROPIA (ISORROPIA II: Nenes et al. (2007); Fountoukis and Nenes (2007)) is used to calculate SIA concentrations from the compound concentrations in the gaseous and aerosol phases depending on the ambient conditions (temperature and humidity). The secondary organic aerosols from anthropogenic sources are treated in MOCAGE with their emissions scaled to primary anthropogenic organic carbon emissions (Descheemaecker et al., 2019). The scaling factor is derived from aerosol composition measurements (Castro et al., 1999). All types of aerosols use the same set of 6 cross-sectional size bins, ranging from $2.0 \times 10^{-3}$ μm to 50 μm with size bin limits of 2, 10, and 100 nm, and 1, 2.5, 10 and 50 μm.





### 3.1.4 Emissions

The surface emissions in MOCAGE come from inventories that emit both natural and anthropogenic sources except for sea salt and desert dust that are dynamically emitted using meteorological parameters (Sič et al., 2015). In the case of "accidental" 200 emissions, such as nuclear accidents or volcanic eruptions, emission fluxes are obtained from other sources. To take into account these "accidental" releases of pollutants in MOCAGE, it is required to define the starting time of the injection, the location of the source, the plume height (top and bottom), the total mass of each of the species emitted and the duration of the emission. Concerning the volcanic eruption, the injection of emission follows an "umbrella" profile that corresponds to the injection of 75% of the emissions in the third top part of the plume (Lamotte et al., 2021). This is to represent the fact that most 205 of the mass injected is in the top part of the plume. For passive emissions, we inject the emission in a 50m-depth layer starting from the volcano vent altitude (Lamotte et al., 2021).

### 3.2 Model improvements

To represent the halogen chemistry in volcanic plumes, several changes and upgrades have been made to the MOCAGE standard version described in Section 3.1. Before explaining these improvements, we present the reactions taking place in 210 volcanic plumes and leading to BrO production, i.e. the bromine-explosion cycle.

### 3.2.1 The bromine-explosion cycle

The following chemical reactions (R1) to (R11) represent the bromine-explosion cycle that occurs in volcanic plume. This cycle has been documented in several previous studies (e.g., Roberts et al., 2014; Jourdain et al., 2016; Gutmann et al., 2018; Surl et al., 2021).

$$HBr + OH \rightarrow Br + H_2O \tag{R1}$$

$$Br + O_3 \rightarrow BrO + O_2 \tag{R2}$$

$$BrO + HO_2 \rightarrow HOBr + O_2 \tag{R3}$$

$$BrO + NO_2 \rightarrow BrONO_2 \tag{R4}$$

$$BrONO_2 + H_2O(aerosols) \rightarrow HOBr + HNO_3 \tag{R5}$$




$$HOBr + HBr(aerosols) \rightarrow Br2 + H_2O \tag{R6}$$

$$HOBr + HCl(aerosols) \rightarrow BrCl + H_2O \tag{R7}$$

$$Br_2 + h\nu \rightarrow 2Br \tag{R8}$$

$$BrCl + hv \rightarrow Br + Cl \tag{R9}$$

$$Br + HCHO \rightarrow HBr + HCO \tag{R10}$$

$$Br + HO_2 \rightarrow HBr + O_2 \tag{R11}$$

The bromine–explosion cycle corresponds to the autocatalytic formation of BrO summarised by the reactions (R12) and (R13).

$$BrO + HO_2 + HBr(aerosols) + 2O_3 \rightarrow 2BrO + 3O_2 + H_2O \tag{R12}$$

$$BrO + NO_2 + HBr(aerosols) + 2O_3 \rightarrow 2BrO + 2O_2 + HNO_3 \tag{R13}$$

The bromine-explosion cycle is fed by the bromine atoms provided by the hydracid HBr. The most important gas phase reaction of HBr is with OH radicals (R1) and allows for the production of Br radicals. Br radicals react with $O_3$ to produce BrO molecules (R2). Then, the BrO molecules react with $HO_2$ and $NO_2$ (R3 and R4 reactions) and produce the new bromine species HOBr and $BrONO_2$. The hydrolysis of $BrONO_2$ produces additional HOBr (R5). In the presence of sulfate aerosols within the volcanic plume, HOBr molecules react competitively either with HBr or HCl to produce $Br_2$ or BrCl (R6 and R7) and produce $Br_2$ or BrCl, respectively, as a function of within-aerosol processes (as discussed by Roberts et al. (2009)). During daytime, $Br_2$ and BrCl molecules are photolysed and produce 2 atoms of Br (R8) or Br and Cl (R9), respectively. The photolysis of $Br_2$ (R8) is particularly important in the bromine-explosion cycle since it provides 2 Br radicals that accelerate the formation of BrO in the plume (R2). This reaction competes with Br reacting with formaldehyde (R10) and with $HO_2$ (R11) to produce



back HBr. At sunset, BrO production decreases and both $Br_2$ and BrCl accumulate as reservoirs of bromine. At night, the cycle stops because there is no photolysis of $Br_2$ and BrCl and consequently no fresh production of Br radicals.

In summary, the cycle leading to the rapid and efficient production of BrO from HBr in the volcanic plume during daytime consists of gas phase and heterogeneous phase reactions. This cycle initiates only with reactive bromine in the presence of acidic aerosols, which enables the possibility for heterogeneous reactions. The chain of reactions can be summarised by the

255 budget reactions (R12) and (R13) whereby 2 molecules of $O_3$ are destroyed in the bromine cycle. Ozone can also be destroyed (or reformed) by other cycles between BrO and different halogen species.

### 3.2.2 New developments of the halogen chemistry scheme

To represent the tropospheric bromine-explosion cycle in volcanic plumes within the MOCAGE 3D model, it is necessary to modify the initial version of the MOCAGE chemistry scheme. We used the chemical scheme developed in the MOCAGE

1D vertical profile version for volcanic halogens (Marécal et al., 2022), which is based on Surl et al. (2021). With reference to modelling and observation studies from the literature, Marécal et al. (2022) showed that this chemical scheme is able to reproduce the bromine-explosion cycle consistently. As in Marécal et al. (2022), we added the $Br_2$ species into the MOCAGE 3D version, its photolysis (R8), the heterogeneous reactions (R6) and (R7) and the hydrolysis of $BrONO_2$ (R5). Other reactions that include chlorine and bromine species to account for tropospheric halogen reactions with volatile organic compounds were

also added. All the halogen reactions introduced are listed in the supplementary material of Marécal et al. (2022).

### 3.2.3 Other new developments

One of the limitations of the work done with the 1D vertical profile version of MOCAGE and presented in Marécal et al. (2022) is that OH and $HO_2$ are treated as a family species and are thus individually represented as diagnostic species. This is why Marécal et al. (2022) could not test the impact of OH volcanic emissions on the halogen plume chemistry, as done

in previous studies (e.g Surl et al., 2021). To overcome this issue and more generally to improve the MOCAGE model, we upgraded MOCAGE to represent OH and $HO_2$ as prognostic species thanks to the implementation of the Rosenbrock solver generated using the Kinetic Pre-Processor (KPP) software (Sandu and Sander, 2006). We use the standard four-stages, third order Rosenbrock solver (Rodas3, Sandu et al. (1997)). This solver is different from the Eulerian Backward Implicit (EBI) solver which had been used until now in MOCAGE. This choice of the third order Rosenbrock solver was motivated by an

improved computation accuracy although at a higher computational cost.

### 3.3 Setup of the simulations

The analysis of the impact of the volcanic halogens is mainly done on a reference experiment simulating the eruption (called "main" hereafter) and on a similar simulation without the volcanic emission giving the background conditions (called "novolc" hereafter). In addition, we ran two sensitivity simulations to assess the impact of using a different composition for the volcanic





emissions. We first describe the general setup common to all simulations that corresponds to the "novolc" experiment, and then the specific features of the 3 simulations that include the volcanic eruption.

### 3.3.1 General configuration

All simulations are run from 24 December 2018 at 00:00 UTC to 31 December 2018 at 00:00 UTC. We use two geographical domains. The global domain has a 2° latitude x 2° longitude horizontal resolution and the regional domain has a 0.2° latitude 285 x 0.2° longitude resolution covering the Mediterranean basin. The global domain provides the boundary conditions to the regional domain.

The meteorological parameters are from ARPEGE analyses, the operational weather forecasting model of Météo-France (Courtier et al., 1991). In the MOCAGE simulations, the main chemical state resulting from the chemistry processing is usually calculated every 15-minutes except for the transport by advection, which is calculated hourly. This was the configuration used 290 by Marécal et al. (2022) but they showed in their MOCAGE 1D vertical profile simulations that this time step is too long to represent the strong bromine-explosion at the very first stage of the volcanic plume (first 1-2 hours). To improve this, we set in our simulations the time steps of all processes to 5 minutes except for the advection which is set to 15 minutes.

The chemical scheme used is that described in section 3 and includes the new developments presented in section 3.2. The configuration of the simulations includes the explicit representation of the aerosols and in particular the sulfate aerosols (Guth 295 et al., 2016). This is different from what was used in Marécal et al. (2022) in which the sulfate distribution was represented by the effective radius of sulfate aerosols (i.e., mean surface area-weighted radius) and not by an explicit distribution of aerosols as in the present study.

The initial conditions for the chemical species and aerosols come from the original MOCAGE version (i.e., without the new developments described in section 3.2). The concentrations of halogen compounds (bromine and chlorine species) in the tro-300 posphere are initialised to zero at the beginning of the simulations for two reasons: (1) the original MOCAGE version does not give reliable tropospheric halogen concentrations since the halogen scheme is designed for the stratosphere only and (2) this allows us to quantify only the effect of the halogens from the volcanic plume.

Regarding the emissions, except for the volcanic eruption emissions, we use the same setup as in Lamotte et al. (2023). The anthropogenic emissions are from the MACCity emissions (Lamarque et al., 2010). The volatile organic compounds (VOCs) 305 are from the MEGAN-MACC inventory (Sindelarova et al., 2014). The biogenic nitrogen oxide ($NO_x$) emissions are derived from the GEIA dataset (Yienger and Levy, 1995) while $NO_x$ from lightning is based on Price et al. (1997) and parametrized using the meteorological parameters. Organic carbon and black carbon sources are taken into account according to MACCity (Lamarque et al., 2010). Oceanic DMS emissions are given by a monthly climatology (Kettle et al., 1999). Finally, the biomass combustion emissions and their associated injection altitudes are from GFAS daily products (Kaiser et al., 2012).

### 3.3.2 Setup of the "main" simulation

To input the volcanic emissions in the accidental release feature of MOCAGE, we use the $SO_2$ emission estimation from the SEVIRI satellite measurements (Corradini et al., 2021) for the Mt Etna Christmas 2018 eruption on the basis of the study of





**Table 1.** Molar ratios to $SO_2$ used in the MOCAGE "main" simulation to set the emissions of the species other than $SO_2$.

| Species | Molar ratio to $SO_2$ |
|---|---|
| $SO_2$ | 1 |
| HCl | 0.44 |
| Br | $1.5 \times 10^{-4}$ |
| HBr | $4.5 \times 10^{-4}$ |
| OH | $1.0 \times 10^{-3}$ |
| NO | $4.5 \times 10^{-4}$ |
| Sulfate aerosols | $2.0 \times 10^{-2}$ |

Lamotte et al. (2023). Lamotte et al. (2023) showed that the SEVIRI estimation gives better results overall than the ground-based FLAME estimates compared to $SO_2$ column concentrations derived from various independent space-borne observations.

The volcanic $SO_2$ emissions used in the "main" simulation are presented in Fig. 1. They are from the SEVIRI estimates when available (i.e., from 24 December at 10:45 UTC to 30 December at 14:00 UTC). Before (from 24 December 00:00 UTC and to 24 December 10:45 UTC) and after (from 30 December 14:00 UTC to 31 December 2018 at 00:00 UTC), $SO_2$ fluxes were set at a constant value with fluxes corresponding to the passive $SO_2$ emissions from the Carn et al. (2017) passive emission inventory (red dashed lines in Fig. 1) as in Lamotte et al. (2023). In the SEVIRI data, there is a measurement gap linked to the

presence of meteorological clouds on 25 December (Corradini et al., 2020, 2021). We filled this gap by assuming a constant $SO_2$ emission flux of 0.43 kt/h (black dashed line in Fig. 1). This value is higher than what was assumed in Lamotte et al. (2023) since their values lead to an underestimation of $SO_2$ concentrations with respect to the satellite-derived $SO_2$ concentrations. The time variation of the emission fluxes for the other species than $SO_2$ are set with respect to the $SO_2$ emission flux shown in Fig. 1. The molar ratios of $SO_2$ to the others species emitted (HBr, HCl, Br radical, OH, NO and sulfate aerosols) are

presented in Table 1. Since there is no measurement close-to-vent of the composition of the eruption, we use the same emission composition as in the modelling study of Mt Etna of Surl et al. (2021). The $HCl/SO_2$ and chlorine-to-bromine ratios are from the comprehensive analysis of measurements gathered between June 2010 and June 2012 at two craters of Mt Etna (Bocca Nova and Northeast craters) that is reported in Wittmer et al. (2014). The emissions include a partition of the total bromine into HBr and Br that represents the production of Br from HBr at very high temperature at the vent and that is based on thermodynamic

modelling estimates from Roberts et al. (2014). We define NO and OH emissions formed from high temperature processing as in Surl et al. (2021). The $sulfate/SO_2$ ratio is derived from the crater-rim aerosol measurement from Roberts et al. (2018).

Regarding the maximum height of the volcanic emission, we used the same as in Lamotte et al. (2023) (Fig. 1). They come from SEVIRI for the maximum injection height (8 km) of the strong emission on 24 December and from visible cameras for the other days. For the $SO_2$ passive emissions before and after the eruption (before 24 December 11:15 UTC and after 30

December 14:00 UTC), we set the maximum height of injection to 3350m, i.e., 50 m above Mt Etna's altitude.





**Table 2.** Volcanic emission rates of species in the different simulations relative to those of the "main" simulation, $Br_{tot}$ = HBr + Br and n/a means not applicable. HBr:Br gives the percentages of HBr and Br with respect to $Br_{tot}$.

| Simulations | Relative volcanic emission rate | | | | | | HBr:Br |
| --- | --- | --- | --- | --- | --- | --- | --- |
| | $SO_2$ | HCl | $Br_{tot}$ | OH | NO | Aerosols | |
| main | 1 | 1 | 1 | 1 | 1 | 1 | 75:25 |
| novolc | 0 | 0 | 0 | 0 | 0 | 0 | n/a |
| stest_ini | 1 | 1 | 1 | 0 | 0 | 0 | 100:0 |
| stest_aero | 1 | 1 | 1 | 1 | 1 | 2 | 75:25 |

### 3.3.3 Setup of the sensitivity simulations

We performed two sensitivity tests to evaluate the impact on the BrO production and associated ozone loss at the regional scale of different chemical compositions of the emissions since there are large uncertainties on high-temperature processing at the vent when magmatic air first mixes with atmospheric air. The choice of the sensitivity experiments was done on the basis of the study of Marécal et al. (2022). We selected their two simulation setups for emission composition giving an important response on BrO concentrations. The "stest_ini" simulation corresponds to the use of the raw emissions not accounting for the change of composition at high temperatures. This means that only $SO_2$, HBr and HCl are emitted (Table 2). In the simulation "stest_aero", we increase the primary sulfate flux by multiplying by 2 the aerosol molar ratio to $SO_2$ (Table 2).

## 4 Results and discussion

### 4.1 Comparison of the "main" simulation with TROPOMI satellite measurements

To assess the representation of the volcanic plume in the "main" simulation, we compare the tropospheric $SO_2$ and BrO columns from MOCAGE (Fig. 3) with those retrieved from the TROPOMI satellite measurements (Fig. 2). The MOCAGE simulations are interpolated at the time and location of the TROPOMI satellite measurements.

Corradini et al. (2021) estimated that the total uncertainty of the $SO_2$ flux is approximately 45%. In addition, the $SO_2$ and BrO columns retrieved from the TROPOMI instrument have uncertainties of the order of 35%. Because of these uncertainties, we can only make a semi-quantitative evaluation based on the comparison of the maps from the model (Fig. 3) and from the observations (Fig. 2).

As in Lamotte et al. (2023), we note a good consistency in terms of the location of the $SO_2$ volcanic plume for the whole eruption period, meaning that the emission height estimation used is good and that the transport of the plume at the regional scale is well represented by the model. However, the plume is more spread out horizontally with less detailed features in the "main" simulation than observed by TROPOMI. This is a consequence of the coarser horizontal resolution of the model compared to the TROPOMI pixel size. The model also cannot represent the small-scale features resulting from very strong



**Figure 3.** Tropospheric column of $SO_2$ (left column) and BrO (right column) in molec.cm$^{-2}$ from the "main" simulation from 25 to 30 December 2018.





wind shears (e.g. on 28 December around 36°N) because of its relatively coarse vertical resolution in the mid and upper troposphere (700 – 800 m). Regarding the $SO_2$ column concentrations, they have generally the same order of magnitude in the

observations and in the simulation, showing that the emission fluxes of $SO_2$ are realistic. The "main" simulation often provides higher column concentrations than TROPOMI but the differences are within the uncertainties of the emission fluxes and of the satellite-derived $SO_2$ columns. Figure 3 also depicts BrO columns from the model. The "main" simulation is able to produce BrO columns of the same order of magnitude as TROPOMI estimates (Fig. 2). This means that the total bromine to $SO_2$ ratio chosen for the emissions is realistic. Also, similarly to TROPOMI, the modelled concentrations of BrO are well correlated with

$SO_2$ concentrations.

To conclude, the comparison between the estimates $SO_2$ and BrO columns retrieved from the TROPOMI satellite measurements and the "main" simulation shows that the transport of the plume is well represented and an overall good consistency for the concentrations knowing the uncertainties in the flux estimations, in the TROPOMI $SO_2$ and BrO columns and in the total bromine/$SO_2$ emission ratio.


## 4.2 Analysis of the evolution of the bromine species from the "main" simulation

In this section, the analysis of the "main" simulation has been done on 3 subdomains to characterise the behaviour of bromine species at different stages of the plume lifetime. Figure 4 shows the 3 subdomains used for the calculations. The first subdomain D1 is close to the volcano and is named "near volcano plume". The second subdomain D2 is further from the volcano and

named "young plume". The third subdomain D3, named the "aged plume", is even further from the volcano than D2. It should be noted that the subdomain D2 does not contain the subdomain D1, and D3 does not contain D1 and D2. Depending on the wind conditions, the plume transitions from D1 to D2 within ∼3 to ∼7 hours, from D2 to D3 from ∼9 to ∼20 hours and from D3 to outside the Mediterranean domain within ∼20 hours for the 24 and 25 December. For the other days, the plume dissipates in D3 before reaching the outer domain boundary.

### 4.2.1 Bromine partitioning

This section analyses the bromine speciation results from the "main" simulation. From the simulated 3D concentration fields, we calculate the tropospheric columns of each bromine species only within the plume by selecting the model grid points for which the tropospheric column of $SO_2$ is greater than $2{\times}10^{15}$ molec.cm$^{-2}$. We then calculate the number of bromine atoms that contributes for each bromine species for these grid points. We finally report each compound as a relative value in percent

with respect to the total number of bromine atoms (Total bromine = HBr+BrO+Br+HOBr+2Br$_2$+BrONO$_2$+BrCl). Since Br$_2$ holds 2 bromine atoms, its columns are multiplied by a factor of 2.

Figure 5 shows the time evolution of the relative partitioning of the bromine species during Mt Etna's Christmas 2018 eruption for the near volcano plume, the young plume, and the aged plume domains. In Fig. 5, the date/time of the first timestep for each of the 3 subdomains is different because the time when the plume enters each domain is different. For the young plume, the

plume starts to enter into the D2 domain from 14:00 UTC and from 20:00 UTC in D3 for the aged plume. Diurnal variations





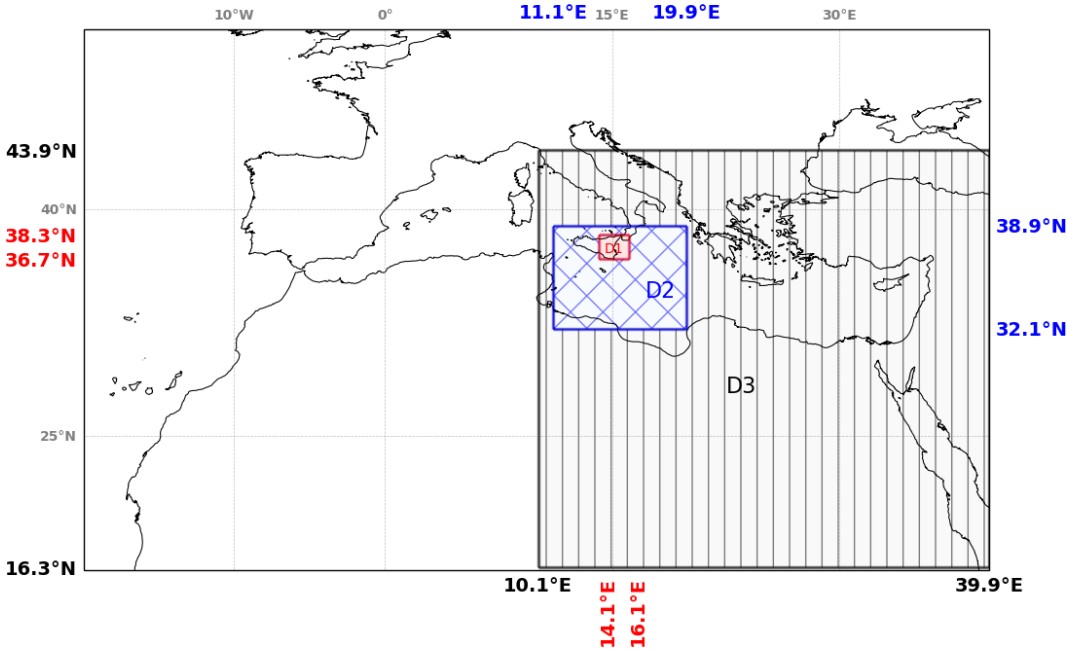

**Figure 4.** Representation of the regional simulation domain over the Mediterranean basin and of the 3 sub-domains used on the analysis of bromine species: D1 for the near volcano plume, D2 for the young plume and D3 for the aged plume.

in halogen composition are evident in each of the domains.

In Fig. 5, the relative quantity of BrO formed during the daytime through the bromine-explosion cycle is different in each of the subdomains, with it being generally more intense in the young plume (Fig. 5b) compared to the near volcano (Fig. 5a) and aged plume (Fig. 5c). There is much greater variability in the apparent strength of the bromine-explosion (visible as the
BrO contribution) in the near volcano plume compared to the other analysis domains. This arises because of the relatively strong contribution from the primary HBr emissions in the near volcano plume and their temporal variability (that scales with the varying $SO_2$ flux). Furthermore, the relatively small size of the D1 domain makes the results in the near volcano plume more sensitive to changes in the meteorological conditions, such as the wind intensity, which leads to accumulation, cloudy conditions that alter the photolysis, and precipitation that leads to the removal of the soluble bromine species (HBr, $BrONO_2$,
HOBr). The causes of this variability and their link with the production of the bromine species is analysed in detail in the next section (section 4.2.2).

Table 3 gives the BrO/HBr ratio in the 3 subdomains at 12:00 UTC. It clearly shows that in the near volcano plume the BrO/HBr ratio is under 1 for most days. This is explained by the strong influence from the primary HBr emissions. However, on 27 and 28 December, this ratio is greater than 1 (3.9 and 3.1). As explained above, this high variability in the D1 domain arises due to its
relatively small surface area, which makes the BrO burdens within the D1 domain very sensitive to meteorological conditions. The column in Table 3 for the young plume shows a larger overall value of the BrO/HBr ratio (3.3 on average) compared to the other two domains but also strong variability from day to day. This variability is linked again to the relatively small domain





**Figure 5.** Time evolution of the relative partitioning of the concentrations of the bromine species from the "main" simulation: (a) from 24 December at 12:00 for the near volcano plume domain, (b) from 24 December at 14:00 UTC for the young plume domain and, (c) from 24 December at 20:00 UTC for the aged plume domain. The method used to calculate the relative partitioning is explained in the text.

size even if larger than the near volcano domain. Finally, the BrO/HBr ratios in the aged plume are lower (2.2 on average) compared to the ratios in the young plume. Overall, this indicates that the bromine-explosion cycle is more efficient in the young plume compared to the aged plume.

At nighttime, the bromine-explosion cycle stops because there are no photolysis reactions. Thus, $Br_2$ and BrCl become the main bromine reservoirs species. Figure 5a shows that the $Br_2$ contribution is larger compared to the contribution from BrCl during all of the nights. As the plume travels further from the vent, the relative contribution of BrCl with respect to $Br_2$ becomes





**Table 3.** BrO to HBr ratio at 12:00 UTC per day for each subdomain (near volcano, young and aged plumes).

| Day/Times (UTC) | BrO/HBr ratio | | |
|---|---|---|---|
| | Near volcano | Young plume | Aged plume |
| 25/12/2018 12:00 | 0.4 | 0.2 | 1.9 |
| 26/12/2018 12:00 | 0.3 | 6.2 | 4.2 |
| 27/12/2018 12:00 | 3.9 | 2.0 | 2.6 |
| 28/12/2018 12:00 | 3.1 | 7.3 | 1.7 |
| 29/12/2018 12:00 | 0.6 | 3.8 | 1.9 |
| 30/12/2018 12:00 | 0.1 | 0.5 | 1.1 |

**Table 4.** $Br_2$ to BrCl ratio at night-time (from 18UTC to 04UTC the day after) per day for each subdomain (near volcano, young and aged plumes).

| Day | $Br_2$/BrCl ratio | | |
|---|---|---|---|
| | Near volcano | Young plume | Aged plume |
| 24-25/12/2018 | 79.5 | 1.8 | 1.5 |
| 25-26/12/2018 | 122.2 | 2.3 | 1.4 |
| 26-27/12/2018 | 62.7 | 1.3 | 0.4 |
| 27-28/12/2018 | 2.5 | 1.3 | 0.4 |
| 28-29/12/2018 | 1.5 | 1.7 | 0.5 |
| 29-30/12/2018 | 2.7 | 1.1 | 0.5 |

more important (Figs. 5b and 5c). This is confirmed by the results presented in Table 4 that gives the average $Br_2$ to BrCl ratio

during the night for the 3 domains. Near the volcano, the $Br_2$/BrCl ratio is very high with an average value of 45.2 for the 6 nights. In contrast, this ratio is much lower in the young and aged plumes on average with ~1.6 and ~0.8, respectively. This is because, as the time from the eruption increases and the plume travels further from the volcano, less HBr is available in the plume to produce $Br_2$ via the heterogeneous reaction (R6), and (R7) becomes more important. We note that in the aged plume, the $Br_2$/BrCl is less than 1 after 26 December. This means that the HOBr + HCl (aerosols) → products (R7) reaction becomes

more efficient relative to the HOBr + HBr (aerosols) → products (R6) reaction because there is not enough HBr available anymore relative to HCl.

HOBr and $BrONO_2$ are also formed in the volcanic plume in secondary chemical reactions alongside BrO, BrCl, $Br_2$, and Br. HOBr and $BrONO_2$ are present both during daytime and nighttime but in greater proportions during daytime and exhibit a nighttime decay as shown in Fig. 5. $BrONO_2$ and HOBr are produced as part of the daytime bromine-explosion cycle from

the chemical reactions (R4) and (R3), respectively. In addition, HOBr is formed from $BrONO_2$ hydrolysis (R5). The relative contribution of both $BrONO_2$ and HOBr compared to the other bromine species increases with the distance/time to the source





(i.e., from the near volcano to the young plume, and from the young plume to the aged plume).

Finally, we note that throughout Fig. 5 HOBr and BrO are generally the dominant species during daytime compared to the other bromine species. These high relative columns of HOBr and BrO, which are indicative of the overall conversion of HBr
to reactive bromine species via R6, are consistent with the results of Surl et al. (2021) and Roberts et al. (2014) after their simulated plumes have aged for 60 or more minutes.

### 4.2.2 Analysis of the chemical reaction rates within the volcanic plume

This section discusses the reaction rates for the production and loss of bromine compounds (BrO, Br, $Br_2$, BrCl, HOBr,
$BrONO_2$ and HBr) from the "main" simulation. We perform this analysis in order to look in more detail at the results discussed in Sect. 4.2.1 that showed the overall bromine partitioning and the BrO/HBr and $Br_2$/BrCl ratios. The objective is to identify and describe the most important chemical processes involving bromine chemistry within the plume. This provides important context to help understand the spatio-temporal evolution of the partitioning and of the ozone loss at the regional scale.

We calculated the time series for the production and loss rates for each bromine species in molec.cm$^{-2}$.s$^{-1}$ units for the 3
subdomains, only for the model grid points within the plume as in section 4.2.1. All of the figures reported in sub-section 4.2.2 (for the production and loss rates of bromine species and molecule number for each bromine species) are for the near volcano plume. The comparison with the young and aged plumes is presented in this sub-section, but all the figures for reaction rates on the young and aged plumes are reported in the Supplement. In addition, since the loss rate of BrO and the production rate of Br include a lot of reactions and many of them make small contributions and can be difficult to see, we also provide in the
Supplement the same figures but without the two main contributions (without red and blue colours) for clarity.

We firstly discuss BrO and Br because they represent the ultimate manifestation of the bromine-explosion and are the species involved and responsible for ozone's destruction. BrO and Br both result from a complex chain of chemical reactions, and we discuss each compound involved in the chain in sequence afterwards, i.e., secondly $Br_2$ and BrCl, then HOBr, HBr, and then finally $BrONO_2$.

– BrO and Br

The reactions rates for BrO and Br for the near volcano plume are reported in Figs. 6 and 7. They show that the production and loss rates for BrO and Br are on the same order of magnitude and that both species are closely coupled and are in equilibrium. The equilibrium and rapid cycling between BrO and Br is mediated by the formation of BrO on the one hand via the $Br + O_3$ reaction and its destruction, which can proceed via several reactions (Fig. 6c and S1) to reform Br or its precursors $Br_2$ and
BrCl. Figure 6b clearly shows that BrO is mainly produced by the $Br + O_3$ reaction compared to the extremely minor reactions: HOBr + O$^{3P}$ and the $BrONO_2$ thermal decomposition reaction.

The BrO destruction reactions (Figs. 6c and S1) bear further note since they can either result in net ozone destruction (BrO + BrO, BrO + ClO, and BrO + O3P), the formation of ozone precursors bearing odd oxygen (BrO + OH, BrO + NO, BrO + $CH_3O_2$), or lead directly to ozone's immediate precursor, O3P, via BrO photolysis. To help further highlight the role of these





three pathways Table 5 shows the relative contribution made by each group of reactions that leads to BrO loss, which has significance for the analysis of the regional impact of the volcanic emissions (see Sect. 4.3.1). Figure 6 and Table 5 highlight that BrO photolysis (in red colour) leads to no net change in ozone but makes a significant contribution to the loss for BrO in the near volcano plume (between 38%-73% and a 47.2% mean). The BrO loss reactions that lead to net destruction of ozone also make a significant contribution to BrO loss in the near volcano plume (1%-52 % and a 45.9% mean) with the reactions

here listed in order of importance (with reference to their corresponding colour in Figs. 6c and S1): $BrO + BrO \rightarrow$ products (blue), $BrO + BrO \rightarrow$ products (black), the $BrO + ClO \rightarrow$ products (cyan), $BrO + ClO \rightarrow$ products (green) and $BrO + ClO \rightarrow$ products (dark brown).

For the near volcano plume, the ozone-destroying reactions are particularly important (45%-52%) in the earlier days of the eruption, 25 to 27 December, compared to the latter three days of the simulation (1%-34%), 28 to 30 December. This reflects

the fact that the ozone-destroying reactions are $XO + XO$ reactions (X = Br or Cl), which rely on the higher concentrations of XO species within the early phases of the eruption and plume-phase. The ozone precursor forming reactions represent the group with the smallest contribution (2%-17% and a 3.4% mean) to BrO loss with $BrO + NO$ (magenta) being the most significant and with $BrO + CH_3O_2$ (orange) playing a more minor role. The final two reactions forming 'Category IV' involving BrO loss are $BrO + HO_2 \rightarrow HOBr + O_2$ and $BrO + NO_2 \rightarrow BrONO_2$ contribute 3%-9% of the BrO loss (a mean of 3.6% ).

Both reactions lead to the formation of secondary reactive bromine species. In particular, the $BrO + HO_2$ reaction leading to HOBr formation is a key reaction within the overall context of the bromine-explosion since HOBr plays a critical role in the continuation of $Br_2$ and BrCl production. Figure 6a shows that the concentration of BrO increases at sunrise for each day reaching its maximum generally around 13:00 UTC, which represents the diurnally driven bromine-explosion cycle. On the contrary, at sunset, the BrO concentrations decrease close to zero because there is no photolysis of both $Br_2$ and BrCl.

Meanwhile, the BrO self-reaction remains active during the nighttime, which results in BrO's rapid depletion.

The production and loss of the BrO are very efficient on 27 and 28 December compared to 24, 25, and 26 December. This difference is mainly due to the reduction of the wind speed/horizontal advection (red line Fig. 6a), which leads to increased accumulation inside the near volcano domain (D1) and allows more time for the bromine-explosion to proceed and reach the peak BrO levels seen. This is consistent with the BrO/HBr ratio presented in Table 3 for the near volcano domain. In addi-

tion, the presence of the precipitation on 25 December leads to some depletion of HBr by wet deposition, which impacts the bromine-explosion efficiency on that day and explains the low production of BrO on 25 and 26 December. From 29 December onwards, the production of BrO decreases linked to the lower emission of $SO_2$ (Fig. 1) which implies a lower emission of HBr. We now discuss the production and loss of Br. Since BrO and Br are so tightly coupled and many of the reactions involving Br are also linked to BrO production and loss much of the description of Figs. 7 and S2 represents a tautology of the previous

descriptions of Figs. 6 and S1. For this reason we limit the discussion here to avoid overlapping descriptions.

Essentially, the primary loss reaction of $Br + O_3 \rightarrow$ products (blue colour in Fig. 6c) is balanced with the BrO photolysis (red colour in Fig. 6b), $BrO + BrO \rightarrow Br + Br + O_2$ (blue colour), $BrO + ClO \rightarrow OClO + Br$ (cyan colour), $BrO + ClO \rightarrow Br + Cl + O_2$ (brown darker colour) and as well as the $BrO + NO \rightarrow NO_2 + Br$ (pink colour) reaction. These reactions recycling BrO and Br constitute a null cycle keeping both species in equilibrium, and the net production of Br therefore results from the



**Figure 6.** Time evolution of : (a) BrO column concentration (blue line) in [molec.cm$^{-2}$] and the average horizontal wind (red line) in m.s$^{-1}$, (b) production rates and (c) loss rates for BrO both in [molec.cm$^{-2}$.s$^{-1}$] from 24 December at 12:00 UTC to 31 December 2018 at 00:00 UTC in the near volcano plume.

reactions producing Br that do not include BrO. This net production is given mainly by the photolysis of BrCl and Br$_2$. Figures 7b and S2 shows that the photolysis of Br$_2$ is the most important reaction for this net production, and that it is a key step at





**Figure 7.** Time evolution of : (a) Br column concentration in [molec.cm$^{-2}$], (b) production rates and (c) loss rates for Br both in [molec.cm$^{-2}$.s$^{-1}$] from 24 December at 12:00 UTC to 31 December 2018 at 00:00 UTC in the near volcano plume.

the beginning of the bromine-explosion cycle. The photolysis of HOBr and BrONO$_2$ do not lead to net Br or BrO formation as each represents a special case. BrONO$_2$ is a temporary daytime reservoir for Br that is formed from the reaction of BrO + NO$_2$ and HOBr is formed primarily from the reaction of BrO + HO$_2$.





**Table 5.** The relative percentage to BrO destruction made by the various classes of BrO reactions. The four different reaction groupings are ozone reforming (BrO photolysis), ozone precursor reforming (BrO + OH, BrO + NO, BrO + CH$_3$O$_2$), ozone-destroying (BrO + BrO, BrO + ClO, and BrO + O$^{3P}$), and grouping four that includes two reactions that form secondary reactive bromine species (BrO + HO$_2$ and BrO + NO$_2$). Note that in the case of the BrO self-reactions, these count twice towards their percentage contribution of BrO loss. The percentages are shown for each subdomain (near volcano, young plume, and aged plume) for the days 25 to 30 December, on each day for the hours from 7 to 15 UTC (i.e. during daytime).

| Space | Category | \ Fraction per day | | | | | | |
|---|---|---|---|---|---|---|---|---|
| | | 25/12 | 26/12 | 27/12 | 28/12 | 29/12 | 30/12 | Average of the 6 days |
| Near Volcano | Ozone reforming | 38% | 42% | 43% | 60% | 63% | 73% | 47.2% |
| | Ozone precursor reforming | 6% | 7% | 2% | 3% | 5% | 17% | 3.4% |
| | Ozone destroying | 52% | 45% | 52% | 34% | 26% | 1% | 45.9% |
| | IV | 4% | 6% | 3% | 3% | 6% | 9% | 3.6% |
| Young plume | Ozone reforming | 39% | 55% | 61% | 58% | 65% | 76% | 57.0% |
| | Ozone precursor reforming | 14% | 7% | 6% | 3% | 2% | 8% | 3.9% |
| | Ozone destroying | 39% | 34% | 29% | 36% | 29% | 6% | 35.3% |
| | IV | 8% | 4% | 5% | 3% | 3% | 10% | 3.7% |
| Aged plume | Ozone reforming | 81% | 64% | 72% | 78% | 76% | 79% | 68.3% |
| | Ozone precursor reforming | 5% | 13% | 10% | 8% | 6% | 8% | 7.1% |
| | Ozone destroying | 8% | 16% | 10% | 5% | 11% | 4% | 18.3% |
| | IV | 6% | 7% | 8% | 8% | 6% | 9% | 6.3% |

Figures 7b and 7c show that Br production and loss occur mainly during the daytime because at nighttime there is no photolysis and BrO concentrations are low. The high Br on 27 and 28 December arises due to the strong bromine-explosion on these days linked to the lower wind speeds and a higher accumulation in the near volcano plume.

By comparing Figs. 6 and S1 to Figs. S3, S4, S5 and S6 in the Supplement, we see that the column concentrations of BrO in the near volcano plume are greater than in the young and aged plumes. Figures S3b and S5b also show that there is a decrease

in the production rates of BrO in the young and aged plumes relative to the near volcano plume (Fig 6b), and this is closely coupled to the concentrations of BrO and other bromine species in the three different plume-domains. This difference is linked to the dilution process, because at points further from the volcano, the plume is more mixed with the ambient air. The increased dilution of bromine, and specifically BrO, further from the volcano also has another impact on the chemistry within the plume. For instance, if we compare the importance of the dominant reaction leading to the destruction of BrO (BrO photolysis in red

colour) in Fig. 6c and Figs. S3c and S5c, we see that the BrO photolysis reaction becomes much more important relative to the XO + XO reactions (X= Br or Cl) under the more dilute plume conditions further from the volcano. These effects likely arise because the halogen oxide self-reactions are second order and therefore follow a square law reaction rate dependence on halogen concentration whereby a reduction in half of concentration would lead to an overall four times slower reaction rate.





This is critical for the wider impact on the troposphere because this represents a shift in importance from the ozone-destroying

XO + XO reactions to the ozone-reforming, BrO photolysis reaction as shown in Table 5. Indeed, the ozone reforming reaction ranges from 38%-73% (47.2% mean) in the near volcano plume, from 39%-76% (57% mean) in the young plume, and from 64%-81% (68.3%) in the aged plume. This implies that as the plume gets further from the volcano the bromine chemistry has a decreasing impact on net ozone depletion. We shall explore the wider impact of this in 4.3.1.

Another notable difference between the three domains is that on 26 December, before 12:00UTC, BrO production is slightly

higher in the young (Fig. S3b) and aged (Fig. S5b) plumes compared to the plume near the volcano (Fig. 6b) during this limited time period. The BrO production is higher in the young plume because there is more volcanic emission this day combined with more advection out of the D1 grid into the D2 grid (this is the opposite of the situation on 27 December).

– $Br_2$ and BrCl

The production and loss rates as well as the concentration of $Br_2$ and BrCl are reported in Fig. 8 and Fig. 9 for the near volcano plume. These two species both form a key step in the bromine-explosion cycle prior to BrO formation. As a reminder, during the daytime, these species are photolysed reproducing Br. In contrast, during the nighttime, these species constitute the primary bromine reservoir compounds due to a lack of photolysis. The key contributors to the production of $Br_2$, and eventually BrCl, are the heterogeneous reactions (R6) and (R7), which are active all day while concentrations of HOBr, HBr and HCl persist.

During daytime, $Br_2$ is produced mainly by the BrO + BrO (blue colour in Fig. 8b) reaction but this is balanced by the loss driven by $Br_2$ photolysis (blue colour in Fig. 8c), which only represents a null cycle. The net production of $Br_2$ only occurs during nighttime and results from the HOBr + HBr (aerosols) → products (R6) reaction (red colour in Fig. 8b).

BrCl has similar overall behaviour to $Br_2$ with the heterogeneous reaction (R7) (red colour in Fig. 9b) being the only reaction leading to the net production of BrCl and only during the night. Similarly to $Br_2$, the null cycling between BrCl and BrO via

(R9) and (R2) makes no net contribution to the production of BrCl. This is despite the BrO + ClO reaction (blue colour in Fig. 9b) making a larger gross contribution to BrCl production during the daytime.

Figures 8b and 9b show that the production of $Br_2$ in the near volcano plume varies relative to the production of BrCl. This is because the heterogeneous reactions HOBr + HBr (aerosols) → products (R6) and HOBr + HCl (aerosols) → products (R7) are in competition. Roberts et al. (2009) explain that HOBr preferentially reacts with HBr over HCl, leading to $Br_2$ being the

main species produced by the heterogeneous reaction. The reaction with HCl leading to BrCl formation is only favoured when the HBr/HCl ratio is considerably smaller (> 100–1000 times lower than the magmatic HBr/HCl ratio). Indeed, one can see the impact of HBr abundance on the $Br_2$ production and column concentrations (Fig. 8a and 8b). It is instructive to compare Figs. 8 and 9 to Figs S11, S12, S13, and S14 in the Supplement that show $Br_2$ and BrCl concentrations and production and loss rates in the young and aged plumes. From this comparison, it is clear that as the plume moves further from the volcano, HBr

is depleted (also consistent with Fig. 5), and the HBr/HCl ratio shifts to the extent that BrCl production comes to dominate over $Br_2$ production. We also see from this that there is a decrease in the overall magnitude of the reaction rates involving $Br_2$ and BrCl as the plume increases in age from near volcano, to young plume, and out to the aged plume. This is reflective





**Figure 8.** Time evolution of : (a) $Br_2$ column concentration in [molec.cm$^{-2}$], (b) production rates and (c) loss rates for $Br_2$ both in [molec.cm$^{-2}$.s$^{-1}$] from 24 December at 12:00 UTC to 31 December 2018 at 00:00 UTC in the near volcano plume.

of the dilution process leading to an overall reduction in the intensity of the bromine chemistry and ozone depletion in the plume. Surl et al. (2021) is the only study that performs a similar analysis as shown in Table 5 looking at role of different XO + XO reactions in the early plume. They report on the importance of these XO + XO reactions in the early plume for ozone







**Figure 9.** Time evolution of : (a) BrCl column concentration in [molec.cm$^{-2}$], (b) production rates and (c) loss rates for BrCl both in [molec.cm$^{-2}$.s$^{-1}$] from 24 December at 12:00 to 31 December 2018 at 00:00 UTC in the near volcano plume.

destruction and that their importance diminishes. Furthermore, they report on a link between wind-driven dilution of the plume and the role this plays in controlling reactions between halogen species, which can be considered an analogy to the volcanic plume dilution and dissipation simulated in this study.





Figure 9a clearly shows that BrCl concentrations are relatively high on the nighttime of 27/28 December, which again links to
the reduction in wind speed and horizontal advection (Figs. 6a) and, consequently, the increase in accumulation within the near
volcano domain. Table 4 shows reduced $Br_2$/BrCl ratios for the near volcano during the nighttime on 27/28, 28/29, and 29/30
compared to the other days. Comparing Fig. 9a and Fig. 8a, BrCl concentrations are much closer to those of $Br_2$ on these days.

– HOBr

HOBr is a bromine species formed in the volcanic plume like BrO after the primary emission of HBr. This species participates
in the key bromine-explosion cycle by reacting both with the HBr and HCl species via the heterogeneous reactions (R6) and
(R7) leading to $Br_2$ and BrCl, respectively. Thus, the discussion of HOBr follows naturally after the discussion of $Br_2$ and
BrCl.

Figure 5a shows that HOBr (yellow colour) is available both during the daytime and nighttime, but in much smaller quantities
at night. Figure 10b (in daytime) shows that HOBr is mainly produced by the $BrO + HO_2 \rightarrow$ products reaction (red colour). The
production of both BrO and $HO_2$ is dependent on the presence of sunlight, which is why this reaction is only present during
daytime and also why HOBr concentrations are higher during the day. Both the production via the hydrolysis of $BrONO_2$
(blue colour) and via the reaction of BrO with $CH_3O_2$ (green colour) contribute in smaller proportions. The production of
HOBr from both the $BrO + CH_3O_2 \rightarrow$ products and $BrO + HO_2 \rightarrow$ products reactions is mainly related to the abundance and
production of BrO. The direct effect of this can be seen on 27 and 28 December when, according to Fig. 6b, there is a high
production of BrO. Because $BrONO_2$ is available during portions of the night, as shown in the Fig. 5a, $BrONO_2$ hydrolysis
remains the only available means of production for HOBr at night, which is why the concentration of HOBr at nighttime is not
negligible. The residual nighttime levels of HOBr play an important role in maintaining the bromine cycle during the night. The
production of HOBr is greater on 25 and 27 December compared to the other days (Fig. 10b), which is linked to a combination
of increased emission on 25 December and reduced advection, increased accumulation, and longer residence times in the near
volcano domain on 27 December.

The HOBr photolysis (in blue colour) is the main reaction leading to the loss of HOBr (Fig. 10c) during the daytime. The (R6)
(in red colour) and (R7) (in green colour) reactions contribute to HOBr loss to a more minor degree during the day but are the
only loss reactions active at night and critically these reactions support the continuation of the bromine cycle. In general, the
loss of HOBr from the (R7) reaction is favoured during the afternoon compared to the (R6) reaction, e.g., on 25, 27, and 28
December, which is caused by the reduction of the HBr to HCl ratio as HBr is consumed during the day (Figs 11a and 11c).
We note that the loss of HOBr is in equilibrium with the sum of the (R2) and (R3) reactions occurring in sequence, and that
this equilibrium is controlled by $HO_2$ availability.

We look now at HOBr in the young and aged plumes (Supplement Figs. S15 and S16). HOBr shows similar behaviour to the
other bromine species as the plume moves further from the volcano, i.e., a decrease in overall concentration and production
and loss rate driven by increasing dilution.







**Figure 10.** Time evolution of : (a) HOBr column concentration in [molec.cm$^{-2}$], (b) production rates and (c) loss rates for HOBr both in [molec.cm$^{-2}$.s$^{-1}$] from 24 December at 12:00 UTC to 31 December 2018 at 00:00 UTC in the near volcano plume.

– HBr

Figure 11 shows the column concentration and the production and loss rates of the HBr species. HBr plays a critical role in
the bromine cycle via the heterogeneous reactions (R6). It is also the primary bromine species emitted during the volcanic





eruption.

The loss of HBr occurs continuously throughout the day and night (Fig. 11c), which is due to the continuous emission of HBr combined with the presence of HOBr even at night. This means that the heterogeneous reaction leading to the formation of $Br_2$ and forming the core of the bromine-explosion can occur continuously. Nevertheless, the loss rate of HBr during the night

(Fig. 11c) is lower than during the day because HOBr is more abundant during the day as its production depends mainly on the intensity of light for the formation of $HO_2$ and BrO. Generally, this leads to higher levels of HBr at night than during the day. For chemical production (Fig. 11b) excluding the direct emission, HBr is mainly produced by the reaction of Br with HCHO (red colour) and by a small amount from the reaction of Br with $HO_2$ (blue colour). Figure 11b shows that the HBr production is higher on 27 December compared to the other days. The higher rates of HBr production this day are linked to the previously

mentioned effects caused by lower wind speeds in the near volcano domain this day.

Comparing Fig. 11 and Figs. S17 and S18 highlights the depletion of HBr that occurs as the plume moves further from the volcano. Given HBr's role as the primary gas emitted by the volcano fuelling the bromine-explosion, this leads to an overall reduction in its intensity in the regions further from Mt. Etna. Roberts et al. (2014) highlight the importance of the concentration level of the primary bromine source gases in determining the strength of the bromine-explosion and the efficiency of the overall

conversion from HBr to BrO, which in the case of this study can be considered analogous to the effects arising from the depletion and dilution of HBr as the plume travels further from the volcano.

- $BrONO_2$

$BrONO_2$ is a secondary bromine species formed from the bromine-explosion cycle that is present as a temporary reservoir in

the volcanic plume during both the daytime and nighttime (as a larger reservoir during the daytime). Its concentration is lower compared to HOBr and it plays a more minor role compared to the bromine chemistry overall, which is why we look at this species last. Figure 12 shows the concentration and the loss and production rates of the $BrONO_2$ species. The production of $BrONO_2$ occurs only during the daytime (see Fig. 12 b) because its only means of production, $BrO + NO_2 \rightarrow BrONO_2$, relies on the presence of BrO, which is not formed at all in the nighttime.

We also note in Fig. 12b that the production of $BrONO_2$ is very efficient on 25 December compared to the other days, this is related to higher levels of $NO_2$ in the plume during the earlier phases of the eruption, i.e., from 24 December to 26 December. The total burden of $NO_2$ within the near volcano plume is shown in Figs. S21 in the Supplement. There are 3 reactions that lead to the loss of $BrONO_2$ (Fig. 12c): its photolysis (blue colour), the dissociation reaction (green colour) and its hydrolysis (red colour). Only the photolysis and hydrolysis of $BrONO_2$ lead to its loss as shown in Fig. 12c. The $BrONO_2$ dissociation

reaction has no discernible impact. In Fig. 12c, the hydrolysis of $BrONO_2$ dominates during the first 2 days while from 27 December onwards, it is the photolysis of $BrONO_2$ that becomes dominant compared to its hydrolysis. This timing variation is probably due to the presence of clouds in the meteorological forcing on 25 December that impacted the efficiency of the hydrolysis of $BrONO_2$.

Comparing Fig. 12 and Figs. S19 and S20, $BrONO_2$ shows similar behaviour to the other bromine species as the plume moves





**Figure 11.** Time evolution of : (a) HBr column concentration in [molec.cm$^{-2}$], (b) production rates and (c) loss rates for HBr both in [molec.cm$^{-2}$.s$^{-1}$] from 24 December at 12:00 UTC to 31 December 2018 at 00:00 UTC in the near volcano plume.

further from the volcano, i.e., a decrease in overall concentration and production and loss rates driven by increasing dilution.





**Figure 12.** Time evolution of : (a) BrONO$_2$ column concentration in [molec.cm$^{-2}$], (b) production rates and (c) loss rates for BrONO$_2$ both in [molec.cm$^{-2}$.s$^{-1}$] from 24 December at 12:00 UTC to 31 December 2018 at 00:00 UTC in the near volcano plume.





### 4.3 The regional impact of Mt Etna's volcanic emissions

#### 4.3.1 Impact of the bromine production on $O_3$ and OH oxidants and on $CH_4$

Several studies (e.g. Roberts et al., 2009; Jourdain et al., 2016; Surl et al., 2021) have shown that the volcanogenic bromine-
explosion cycle has an impact on tropospheric oxidants, e.g., $O_3$ and OH. As a consequence of these perturbations of atmo-
spheric oxidants, bromine emitted from volcanoes can have an indirect impact on the lifetime of $CH_4$, as was shown in the
modelling study of Jourdain et al. (2016). In this subsection, we therefore focus on the impact of the rapid production of BrO
on the tropospheric burden of $O_3$, OH and $CH_4$ within our modelling domain in the Mediterranean basin.

We first illustrate the impact of BrO production on $O_3$ as the plume travels within the model domain. Figure 13 shows the dif-
ference between the "main" and the "novolc' simulations for the tropospheric columns of BrO and $O_3$ on 26 and 27 December
at 14:00 UTC. BrO columns have large concentrations close to the vent that decrease as the plume travels and dilutes with
background air. Still, BrO exhibits significant concentrations several thousand kilometres from the vent (see for instance the
plume on 26 December over Turkey). Associated with these elevated levels of BrO, there is a strong $O_3$ depletion all along
the plume. The rectangle highlights the location of the part of the plume linked to the high emissions of 26 December before
8 UTC, called Plume26 hereafter. The BrO column reaches very high values in Plume26 on 26 December at 14 UTC. On 27
December, BrO re-forms during daytime from $Br_2$ and BrCl night-time reservoirs in Plume26, but because of diffusion during
transport, the maximum of BrO columns is lower than on 26 December. In Plume26, the $O_3$ depletion reaches about $6.5 \times 10^{16}$
molec.cm$^{-2}$ (2.7 DU) on both days even though the BrO column is lower on 27 December. This is because during daytime on
27 December the formation of BrO occurs for the second time in Plume26 leading to further $O_3$ depletion on 27 December in
addition to the depletion from 26 December. In Plume 26, the reduction of ozone is very strong with a 50 ppb depletion (not
shown) reached at one grid point on 26 December and an apparent lower reduction of 20 ppb reached on 27 December (not
shown), but this is spread over several grid points because of the dilution of the plume during transport. These reductions of
ozone are large in absolute , yet ozone does not completely deplete because of the high background ozone (about 80 ppb) due
to a previous stratospheric intrusion in the Mediterranean basin.

We also consider the impact that changes in BrO-loss chemistry have on ozone loss as the plume travels further from the
volcano. The different means by which BrO loss takes place can either confirm the loss of ozone (e.g., BrO self-reactions),
or lead to the reformation of ozone by BrO photolysis. Table 5 shows a decreasing trend in the ozone destruction reactions as
the plume moves further from the volcano. Meanwhile, the contribution of the ozone-reforming BrO photolysis increases with
distance and time from the volcano. Furthermore, the overall burden of BrO in the near volcano plume (Fig. 6) is several times
higher than in the young and aged plumes (Figs. S3 and S5). This implies that the spatial patterns of ozone loss occurring far
from the volcano in Fig. 13 result to a large extent from ozone depletion occurring near the volcano that is then transported as
an ozone deficit further afield and to a lesser extent to the ozone loss through re-formation of BrO on the days after.

To quantify the overall regional impact of the eruption, Fig. 14 shows the temporal evolution of the total burden of OH,
$O_3$ and $CH_4$ species within the Mediterranean basin. To quantify the effect of the eruption, the tropospheric burdens of OH
(molecules), $O_3$ (kg) and $CH_4$ (kg) are calculated by integrating the difference between the "main" (eruption conditions) and



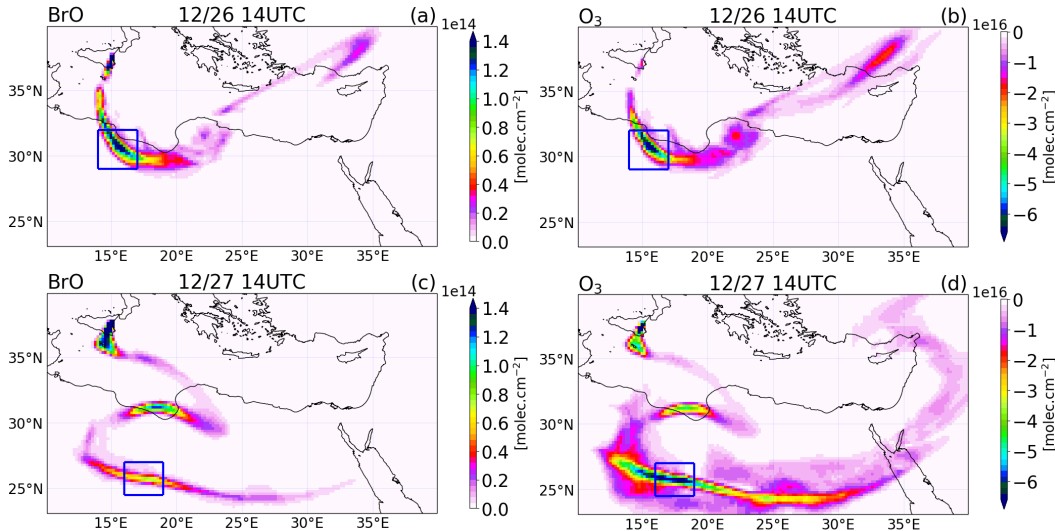

**Figure 13.** Differences between the "main" and the "novolc" simulation of the tropospheric column of: (a) and (c) BrO in molec.cm$^{-2}$, (b) and (d) $O_3$ in DU at 14:00 UTC on 26 (top) and 27 (bottom) December 2018

"novolc" (background conditions) simulations (Table 2) over every tropospheric grid cell in the horizontal and vertical of the whole simulation domain.

The chemistry in the volcanic plume results in a direct reduction of the concentrations of the oxidant OH (maximum during daytime at a peak of around 10:00 UTC on 29 December). This is because it is used in the oxidation reactions of $SO_2$, HCl and HBr (R1). Also, the chemistry of the volcanic plume has a direct impact on the concentration of $O_3$ as presented in this study and by several observations and modelling studies already published. The decrease of $O_3$ leads to a decrease of OH concentration indirectly by the reactions: $O_3 + h\nu \rightarrow O^1D$ and $O^1D + H_2O \rightarrow 2\ OH$. Consistently with the analysis of Fig. 13, there is an additional decrease of the ozone burden (Fig. 14b) from one day to the next linked to daytime re-formation of BrO from its night reservoirs $Br_2$ and BrCl through the bromine-explosion cycle that adds to the BrO formation from emissions of the day.

The reduction of the total mass of OH implies an increase in the lifetime of $CH_4$. Note that this OH reduction dominates over the $CH_4$ loss by reaction with volcanic chlorine as discussed by Jourdain et al. (2016). This is why the total mass of $CH_4$ (Fig. 14c) increases during the period of the simulation. It reaches its maximum around 15:00 UTC on 30 December (around $9.7 \times 10^5$ kg) correlated with the peak of the $O_3$ loss (at approx. $3.5 \times 10^7$ kg). The increase of the $CH_4$ burden ($0.25 \times 10^5$ kg.day$^{-1}$ from 27 December to 30 December) can be compared with estimates of global daily methane emissions based on Saunois et al. (2020) ($1.58 \times 10^9$ kg.day$^{-1}$ on average over the 2008-2017 period). Even if much smaller, it is not negligible considering that we only quantify here the impact of one particular eruption plume over a limited area domain and not of the halogen emissions from all volcanoes at the global scale knowing that some of them like Mt Etna or Ambrym are halogen-rich and have large passive degassing. Therefore at the global scale, volcanic halogen emissions via their effect on OH could have a significant





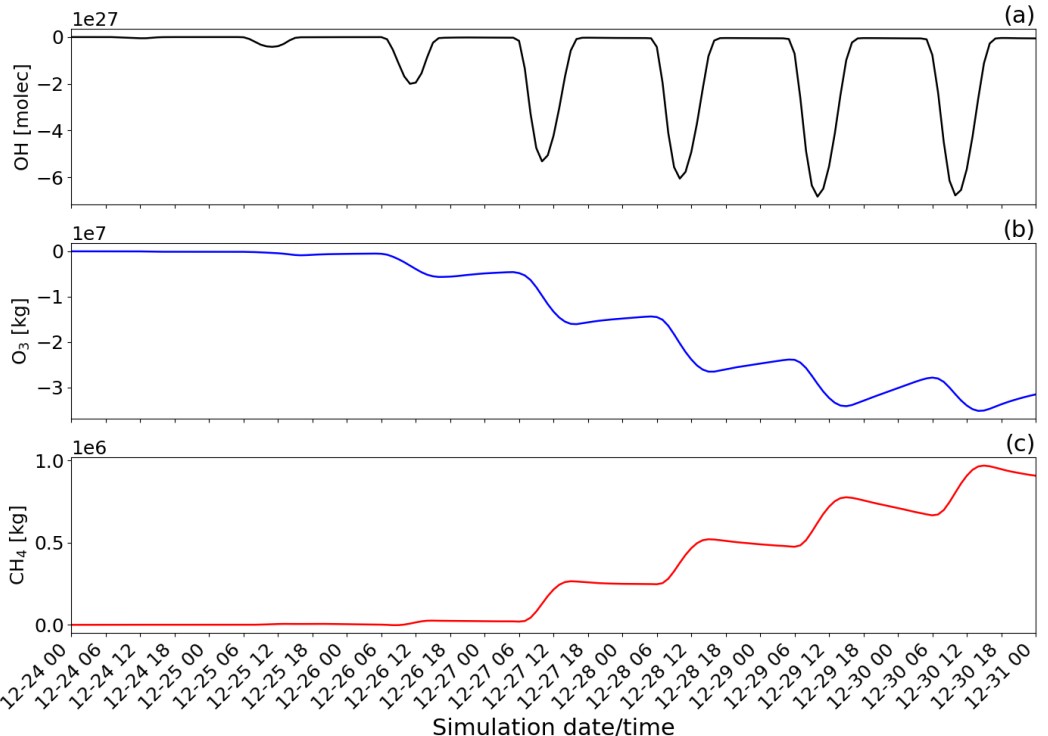

**Figure 14.** Time evolution from 00:00 UTC of 24 December of the difference of the tropospheric burden between the "main" and "novolc" simulations for: (a) OH in [molec], (b) $O_3$ in [kg] and (c) $CH_4$ in [kg] within the whole modelling domain (Mediterranean basin).

indirect impact on the climate since $CH_4$ is among one of the strongest greenhouse gases.

### 4.3.2 Sensitivity studies

Two sensitivity tests were performed. Both tests looked at different scenarios for the chemical composition of the volcanic vent emissions. As shown in Table 2, the "stest_aero" simulation corresponds to the "main" simulation with the only difference
being a doubling of the emissions of primary sulfate aerosols. The other sensitivity test named "stest_init" assumes that there is no high-temperature processing at the vent, i.e., only magmatic emissions of $SO_2$, HBr and HCl are included (Br, OH, NO and sulfate aerosols emissions are turned off). The objective is to evaluate their impact on the BrO production and the associated ozone depletion, knowing that these tests correspond to two extreme sensitivity cases as found in Marécal et al. (2022). They showed with MOCAGE 1D simulations that doubling the primary sulfate emission leads to a more rapid production of BrO
with maximum values increased by ~7% while not taking into account high-temperature processing at the vent strongly reduces and slows down the production of BrO.

Figure 15 shows the time evolution of BrO and $O_3$ concentrations of the "stest_aero" and "stest_ini" sensitivity tests com-



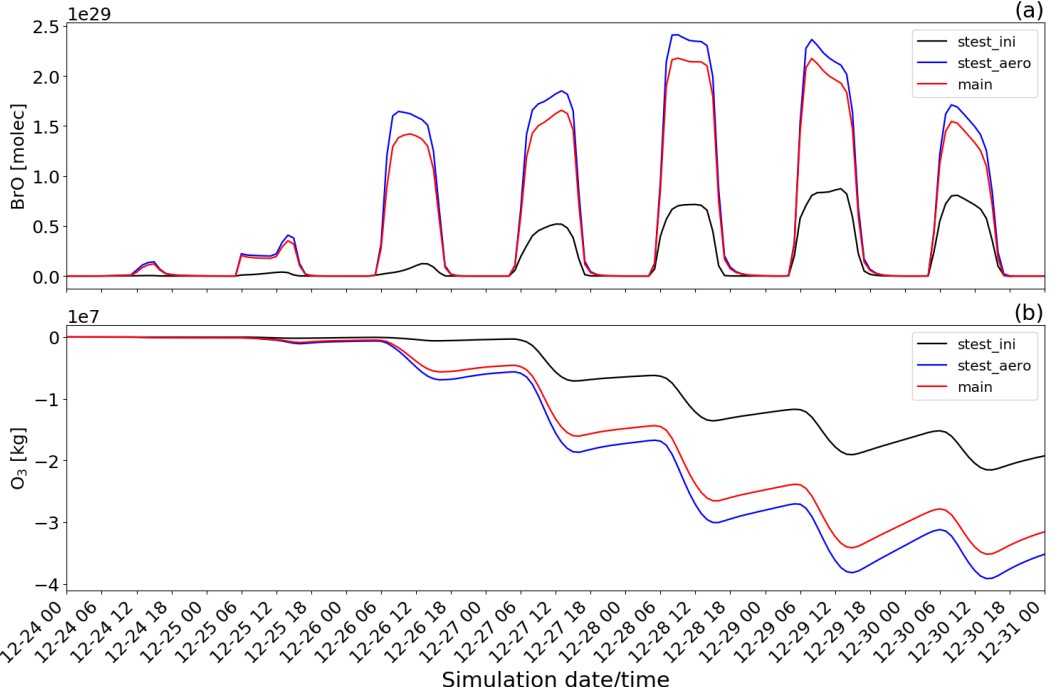

**Figure 15.** Time evolution from 00:00 UTC of 24 December of the: (a) BrO concentration in [molec] and (b) $O_3$ mass in [kg] for "main" minus "novolc" (red line colour), "stest_ini-novolc" (black line colour) and "stest_aero-novolc" (blue line colour) simulations.

pared with the "main" simulation. First, we look at the general behaviour across all three simulations ("main", "stest_aero", "stest_init"). Fig. 15 clearly shows that the temporal evolution of BrO production associated with ozone loss in the two sensitivity tests follows the behaviour in the "main" simulation. However, the BrO and the ozone concentrations vary from one simulation to another.

By doubling the amount of sulfate aerosols in the "stest_aero" sensitivity test, we see that the bromine chemistry is sensitive to the abundance of sulfate aerosols since BrO concentrations increase in the "stest_aero" simulation (maximum ~$2.4 \times 10^{29}$ molecules) compared to the "main" simulation (maximum ~$2.2 \times 10^{29}$ molecules) throughout Fig. 15a. The relative enhancement of BrO between "stest_aero" and "main" varies between 10% to 18% between the different days. The increase in BrO production is related to the increase in sulfate surface area caused by the doubling of sulfate aerosols. This is due to the dependence of the reaction rate on HOBr uptake, which is surface are limited. Due to the highlighted sensitivity, error in simulating the specific surface area of the sulfate aerosol is therefore a source of uncertainty in our simulations. These findings are fully consistent with Marécal et al. (2022) and with Roberts et al. (2014) who show that the reactive absorption of HOBr on halogen-rich aerosols enhances the subsequent transformation of the Br radical to other reactive bromine species. We also note that this high BrO concentration in the "stest_aero" simulation is associated with a significant ozone loss with a maximum of ~$3.9 \times 10^7$ kg. In the "main" simulation the maximum is ~$3.5 \times 10^7$ kg (Fig. 15b). The difference between "stest_aero" and "main" simula-



tions increases with time because there is more BrO production in the stest_aero simulation. At the end of the simulation, the $O_3$ loss is 11% higher in "stest_aero" with respect to the "main" simulation.

The results from "stest_ini" simulation shows that BrO production is much lower (~$8.7 \times 10^{28}$ molecules for the maximum) and also much slower than the "main" simulation. Even if it increases with time, it only reaches 52% of the BrO of the "main" simulation on the last day with a limited impact on ozone depletion (maximum ~$2.2 \times 10^7$ kg). The results of this sensitivity test are consistent with previous studies (e.g. Roberts et al. (2009); Marécal et al. (2022)). This implies that the bromine production cycle needs the Br radical and the primary sulfate in the volcanic emission at the beginning of the simulation to trigger the

rapid onset of the bromine-explosion, and yield BrO consistent with the TROPOMI observations.

## 5    Conclusions

Volcanoes are known to be important emitters of gaseous species and aerosols into the troposphere from passive emissions and moderate volcanic eruptions. In addition to $H_2O$, $CO_2$ and sulfur compounds, halogen hydracids (HBr, HCl) are also emitted. HBr can be converted into reactive bromine in volcanic plumes by the bromine-explosion cycle, which leads to ozone loss.

This cycle has been explored in several modelling studies but mainly in the first hours after emission. Only one study was done on the impact of volcanic halogens at the regional scale for a case of strong passive degassing at Ambryn located in the tropical Pacific region (Vanuatu) (Jourdain et al., 2016).

We present in this paper the results of a new modelling study at the regional scale with the chemistry-transport model MOCAGE for a very different case study. Here we focus on the Mt Etna volcanic eruption that occurred around Christmas 2018 and that

lasted 6 days. This eruption was characterised by emissions reaching 4.5 to 8 km height and strong emissions. The main objectives of this study are to test the ability of the regional 3D CTM MOCAGE model to simulate the bromine-explosion cycle during this case study, to analyse the different chemical processes in the volcanic plume at different distances from the vent and to quantify its impact on the tropospheric composition at the regional scale (i.e. over the whole Mediterranean basin). In order to simulate the chemistry of halogens in volcanic plumes in the MOCAGE 3D model, we implemented the chemical

scheme developed in the MOCAGE 1D version (Marécal et al., 2022) for volcanic halogens, which is based on Surl et al. (2021). Marécal et al. (2022) showed that their chemical scheme was able to reproduce the bromine-explosion cycle in a manner consistent with previous modelling studies and observations. In our study, we set up a "main" simulation that is thoroughly analysed and two sensitivity simulations to the composition of the emissions. All simulations are run from 24 December 2018 at 00:00 UTC to 31 December 2018 at 00:00 UTC using a regional domain covering the Mediterranean basin with a resolution

of $0.2° \times 0.2°$.

The results of the "main" simulation are compared with observations from the TROPOMI satellite showing a good agreement on the location of the plume even when its transport patterns are complex. Regarding the concentrations, the modelled $SO_2$ and BrO column concentrations are generally higher but still consistent with TROPOMI columns taking into account the uncertainties on the emission fluxes (45%) used in the model that were retrieved from SEVIRI satellite observations and the

uncertainties on the satellite retrievals of $SO_2$ and BrO columns from TROPOMI observations (35%) used for the model



comparison.

The analysis of the results is carried out on 3 sub-domains, a first domain close to the vent (near volcano), a second sub-domain further from the vent (young plume) and a third sub-domain which is the furthest from the vent (aged plume). The analysis of the bromine species partitioning provides a detailed picture of the simulated bromine chemistry cycle during the daytime highlighting the role played by all the different bromine reactive gases formed as a result of the bromine-explosion cycle. As expected, BrO is rapidly formed during daytime while $Br_2$ and BrCl are consistently found to be bromine reservoir species at night. The analysis of bromine speciation shows good consistency with the previous modelling studies. The chemical processes for the production and loss of bromine species and associated concentration evolve with time from the initial point of the plume's emission leading to changes from the near volcano to the young plume domain, and from the young plume to the aged plume domain. This is related to the dilution process and the depletion of the primary emitted HBr relative to other chemical components, e.g., HCl. The primary impacts of the plume mixing with the ambient air are an overall less intense bromine chemistry and a shift from the dominance of the XO + XO (X= Cl and Br) halogen oxide self-reactions that mediate ozone depletion, which both lead to decreased ozone depletion as the plume mixes and ages. The rate of the second-order self-reactions of the halogen oxides depends on a square law with respect to the halogen oxide concentrations. The depletion of HBr relative to HCl with plume age causes a shift in the relative concentrations of each halide ion in the sulfate aerosols, which leads to a shift from the dominance of the production of $Br_2$ to BrCl proceeding, respectively, via (R6) and (R7).

We also found that there is a day-to-day variability for all bromine species in all subdomains linked to varying meteorological conditions (wind, cloud and precipitation) and to the time variations of the emission flux. These effects are particularly strong in the near volcano domain because of it is small and it is also where the volcanic emissions are injected.

The simulations showed a large regional impact on oxidants (reduction of OH and ozone), which is caused by bromine and chlorine chemistry in the volcanic plume. In addition to the impact on the oxidants, the bromine and chlorine chemistry have the indirect effect of increasing the burden of methane ($CH_4$). These results are consistent with the plume study of Jourdain et al. (2016) for the Ambrym volcano even though the emissions, background conditions and the model used are not the same. A further step is to estimate the impact of the volcanic halogen emissions from all volcanoes at the global scale, in particular on $CH_4$ which is an important gas for Earth's climate. This will be the subject of a future study with the MOCAGE model using its global configuration.

The sensitivity to the composition of the volcanic emission has been tested by running two simulations. The first one assumes twice more primary sulfate concentrations that leads to an increase of BrO production and more ozone loss. The second one uses raw magmatic emissions without taking into account the high temperature production at the vent of radicals, oxidants and primary sulfate. This simulation provides a strong decrease of BrO production and less ozone depletion. These results are consistent with previous studies and highlight the need of having reliable information on the composition of the volcanic emissions. There are still high uncertainties in the processes occurring at very high temperature when magmatic air first mixes with atmospheric air. There is on-going research on this subject that should reduce the uncertainties regarding the composition of the emissions used in 3D atmospheric chemistry simulations.



*Data availability.* The volcanic $SO_2$ inventory used for passive degassing emissions before and after the eruption is available on Carn et al. (2017) supplementary information (https://doi.org/ 10.1038/srep44095). The results of simulations used in this study are available upon request.

*Author contributions.* HN, PDH and VM designed the study and the model experiments. The model developments and the simulations were carried out by HN, with help from LS, TR, SP, BJ, JG and MB. The emission fluxes of $SO_2$ and injection heights for the eruption were 780 processed and provided by SC, GS and LG. The TROPOMI column of $SO_2$ and BrO were processed and provided by SW and TW. The paper was written by HN, PDH and VM, and commented on and edited by TR, LS, SW, TW, GS, SP, JG and MB.

*Competing interests.* The authors declare that they have no conflict of interest.

*Acknowledgements.* The authors acknowledge funding from the ANR project VOLC-HAL-CLIM (Volcanic Halogens: from Deep Earth to Atmospheric Impacts, ANR-18-CE01-0018) and the Le Studium Consortium H'allo Volcano. The authors also thank Météo-France and 785 the Agence Nationale pour la Recherche for funding Herizo Narivelo's PhD and the Centre National de la Recherche Scientifique for the administration of Herizo Narivelo's contract. Partial funding for Paul David Hamer was provided by NILU under the EO SIS project (B-121004).



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
