# Peer review of "A regional modelling study of halogen chemistry within a volcanic plume of Mt Etna's Christmas 2018 eruption"

_EGUsphere, 2023_

## Author Comment (AC1)

**RC1**: 'Comment on egusphere-2023-184', Anonymous Referee #1, 30 Mar 2023

The authors are thankful to the reviewer for their comments and suggestions. They allowed us to improve the paper by making it clearer. After each of your comments/suggestions in quoted italics you will find the authors' s response to each comment in bold text colored on blue. Where more substantial changes were made to the manuscript, we have quoted these in bold red below. We also have made some minor corrections to spelling and wording and one citation Warnach et al., 2023 (https://egusphere.copernicus.org/preprints/2023/egusphere-2023-933/) to the final verison of this paper.

*"Given the substantive dataset collected and reduced, I recommend provided overarching/broader context of the implications your data – basically, over the course of the Eruption, was the air quality adversely impacted? If so/not, to what extent, quantitatively/qualitatively – especially climate and air quality @ Earth's surface/within the boundary layer in those regions. Moreover, were their any associated human health/ecosystem/build-environment impacts associated w/ the Eruption ?"*

**The eruption height varied between 4 km altitude (min) and 8km altitude (max). Therefore, the eruptive emissions injected SO2 and halogen species into the free troposphere and had no major impact on air pollution at the surface even at the local scale over Sicily. Neither were there significant impacts on commercial aviation from the injection of ash. Thus, there are no impacts on human health, the ecosystem, industry, and the environment.**

*"I also suggest considering juxtaposing Figures 2 and 3 somehow as it would be great to see them next to each other to compare TROPOMI satellite column SO2 and BrO profiles and MOCAGE model simulated column BrO and SO2 profiles."*

**We agree that it is useful to be able to see Figures 2 and 3 together so that they can be compared. It is not possible to merge the two figures into one because they will be too small. However, we will try to get the editor to place the two figures next to each other in the published version.**

*"Also, the degree of congruence appears hard to interpret, especially for BrO as TROPOMI column profiles exhibit a background of BrO throughout the region (w/ slight variability in the region) and 6 days of Eruption while MOCAGE shows slight variability in distinct places w/ no excess BrO background (as shown in the TROPOMI BrO profiles). Atop this, is it possible to quantify the degree of congruence of the MOCAGE model results w/ TROPOMI satellite profiles ?"*

**We have now added Fig. S22 in the supplement of our paper to show the linear regression on a logarithmic scale of BrO from the model and the TROPOMI observations. In this figure, the red line corresponds to the linear regression and the black line represents the 1:1 line. The left column (a) shows the representation of the linear regression inside the volcanic plume and those in the right column (b), show the regression line which takes into account the values of the background outside of the plume. Each row in the panel shows a day from 25/12/2018 to 30/12/2018. We notice through these figures that the regression is not perfect and correlation are low from one**

day to another for both the values of the BrO column inside the volcanic plume and the column that take into account the background.

Indeed, there is variance in both the background and plume, which is mainly caused by the statistical variation of the BrO VCD. For the 25 December, for example, the histogram of all BrO VCDs can best be described by a Gaussian distribution with a mu of $1.3 \times 10^{11}$, i.e., almost 2 orders of magnitude lower than the sigma of $9.73 \times 10^{12}$ molecules.cm$^2$. Thus, a statistical uncertainty of $2 \times 10^{13}$ molecules.cm$^2$ has to be assumed, which is in line with the noise seen in figure 2 and the supplement figure S22. The mentioned systematic variations in the background are estimated to be in the order of $5 \times 10^{12}$ molecules.cm$^2$, which seems to strongly impact the regression and correlation and low simulated column densities.

We realize that this is not reflected in the text and we therefore we rephrase the lines 146-150 as well as adding a short explanation of Fig S22:

"The tropospheric columns of $SO_2$ and BrO retrieved in the volcanic plume from the TROPOMI satellite observations around Christmas 2018 (from 25 to 30 December) obtained using a retrieval algorithm based on the DOAS method (Differential Optical Absorption Spectroscopy) (Hörmann et al., 2013; Warnach, 2022, Warnach et al., 2013) are presented in Fig. 2. The SO2 uncertainty is estimated at 35%. For BrO VCD $<4 \times 10^{13}$ the BrO uncertainty is dominated by the statistical variation of the DOAS column retrieval, which is estimated as $2 \times 10^{13}$ molecules.cm$^2$, based on Warnach et al., 2023. For higher columns the uncertainty is estimated at 35%. Furthermore, systematic biases are estimated in the order of $5 \times 10^{12}$ molecules.cm$^{-2}$ (Warnach et al., 2023). The systematic error component in the TROPOMI satellite observations of BrO apparently leads to relatively high and noisy background columns. For more details, we refer readers to the supplement and to Fig. S22, here it is possible to see that the relatively high systematic error significantly degrades the strength of the correlation between the model and satellite observations, particularly for the lower column densities simulated by the model. The $SO_2$ column for 24 December are not shown because the TROPOMI overpass was very close to the beginning of the eruption and thus only captured the plume on a few pixels."

**Citation**: https://doi.org/10.5194/egusphere-2023-184-RC1

---

## Author Comment (AC2)

**RC2**: 'Comment on egusphere-2023-184', Anonymous Referee #2, 28 Apr 2023

We thank you for your useful comments that helped us improving the paper. Our response is organised as follows. After each of your comments in quoted italics you will find the authors' s response to each comment in bold text colored on blue. Where more substantial changes were made to the manuscript, we have quoted these in bold red below. We also have made some small corrections to spelling and wording as your suggestion in the final publication of our paper and we hope that our response to these comment helps reader.

*"Page 2, last sentence of abstract : I don't know all the results of previous volcanic halogen modeling but the statement that the results are consistent with all previous ones seems a bit strange."*

**This statement is unclear and has been replaced by :**

**"All the results of this modelling study, in particular the rapid formation of BrO, which leads to a significant loss of tropospheric ozone, are consistent with previous studies carried out on the modelling of volcanic halogens."**

*"Page 2, line 31: "in the form of SO2" (there are different sulfur species but no different SO2 species)"*

**Done**

*"Page 2, line 35: "magmatic gases" instead of "magmatic air" pp"*

**Done**

*"Page 11, line 255: Which other cycles are meant here ?"*

**We agree that this sentence is not clear enough. We have added some clarification that ozone destruction in the R12 and R13 cycles is mediated by the BrO + BrO self-reaction. The other destruction reactions involve the BrO + BrO, BrO + ClO, and BrO + O3P reactions. The ozone reforming reactions include pre-cursor reforming (BrO + OH, BrO + NO, BrO + CH₃O₂) and direct ozone reforming (BrO + hv). We propose the following changes to the text:**

**"The ozone destruction in the R12 and R13 chemical cycles is mediated by the BrO + BrO self-reaction that leads to the formation of Br₂ and molecular oxygen. Other variations of these ozone destroying cycles are mediated by BrO + ClO and BrO + O3P. Conversely, other reactions of BrO can lead to ozone reformation (BrO + hv) and the formation of ozone precursors (BrO + OH, BrO + NO, BrO + CH₃O₂)."**

*"Page 11, line 268: "….and are thus individually represented as diagnostic species" ??"*

**This sentence is not clear and has been replaced by:**

**"One of the limitations of the work done with the 1D vertical profile version of MOCAGE and presented in Marécal et al. (2023) was that OH was a diagnostic species. In the standard version of MOCAGE used by Marécal et al. (2023), a chemical family approach is used meaning that HOx (H+OH+HO₂) is the variable explicitly represented and the OH concentration is then diagnosed from HOx assuming a photochemical equilibrium."**

*"Page 13, line 335: Certainly not important and just a detail: In an earlier part of the manuscript the height of Mt Etna is given as 3330 m – 50 m above would mean 3380m."*

**Done**

*"Page 20, line 439: I would not use abbreviations in the main text => molecules cm-2 s-1"*

**Done**

**Citation**: https://doi.org/10.5194/egusphere-2023-184-RC2